# Cell type-specific multi-omics analysis of cocaine use disorder in the human caudate nucleus

Lea Zillich [1,2,3,4,5] ✉, Annasara Artioli[2,3,4], Veronika Pohořalá[6], Eric Zillich [1], Laura Stertz[7], Hanna Belschner[1], Ammar Jabali[2,3,4], Josef Frank [1], Fabian Streit [1,5], Diana Avetyan [1], Maja P. Völker[1], Svenja Müller[1,5], Anita C. Hansson [6], Thomas D. Meyer[7], Marcella Rietschel [1], Julia Ladewig [2,3,4], Rainer Spanagel [5,6], Ana M. M. Oliveira [5,8], Consuelo Walss-Bass[7], Rick E. Bernardi[6], Philipp Koch [2,3,4,5] & Stephanie H. Witt [1,5,9]

Structural and functional alterations in the brain's reward circuitry are present in cocaine use disorder (CocUD), but their molecular underpinnings remain unclear. To investigate these mechanisms, we performed single-nuclei multi-ome profiling on postmortem caudate nucleus tissue from six individuals with CocUD and eight controls. We profiled 30,030 nuclei, identifying 13 cell types including D1- and D2-medium spiny neurons (MSNs) and glial cells. We observed 1485 differentially regulated genes and 10,342 differentially accessible peaks, with alterations in MSNs and astrocytes related to neurotransmitter activity and synapse organization. Gene regulatory network analysis identified transcription factors including *ZEB1* as exhibiting distinct CocUD-specific sub-clusters, activating downstream expression of ion- and calcium-channels in MSNs. Further, *PDE10A* emerged as a potential drug target, showing conserved effects in a rat model. This study highlights cell type-specific molecular alterations in CocUD and provides targets for further investigation, demonstrating the value of multi-omics approaches in addiction research.

Cocaine is one of the most frequently consumed illicit psychostimulants worldwide and poses a significant public health challenge. In the US alone, about 4.8 million people, or 1.7% of the population, used cocaine in the past 12 months and 0.5% are diagnosed with a cocaine use disorder (CocUD)[1]. Cocaine use is associated with a range of negative side effects, most prominently cardiovascular and respiratory complications[2] and cocaine-involved death rates are rising[3,4].

A complex interplay between genetic and environmental factors is assumed to contribute to an individual's susceptibility and resilience to CocUD[5]. In recent years, epigenetics has been increasingly studied as

[1]Department of Genetic Epidemiology in Psychiatry, Central Institute of Mental Health, Medical Faculty Mannheim, Heidelberg University, Mannheim, Germany. [2]Department of Translational Brain Research, Central Institute of Mental Health, Medical Faculty Mannheim, Heidelberg University, Mannheim, Germany. [3]HITBR Hector Institute for Translational Brain Research gGmbH, Mannheim, Germany. [4]German Cancer Research Center (DKFZ), Heidelberg, Germany. [5]German Center for Mental Health (DZPG), partner site Mannheim/Heidelberg/Ulm, Mannheim, Germany. [6]Institute of Psychopharmacology, Central Institute of Mental Health, Medical Faculty Mannheim, Heidelberg University, Mannheim, Germany. [7]Louis A. Faillace, MD, Department of Psychiatry and Behavioral Sciences, McGovern Medical School, University of Texas Health Science Center at Houston, Houston, TX, USA. [8]Department of Molecular and Cellular Cognition Research, Central Institute of Mental Health, Medical Faculty Mannheim, Heidelberg University, Mannheim, Germany. [9]Center for Innovative Psychiatric and Psychotherapeutic Research, Biobank, Central Institute of Mental Health, Medical Faculty Mannheim, Heidelberg University, Mannheim, Germany. ✉e-mail: Lea.Zillich@zi-mannheim.de

an important mechanism for how environmental factors influence the transcription of genetic information, thus altering brain function[6]. However, the role of epigenetic changes and their relevance for altering gene expression levels in brain regions implicated in CocUD remains unclear[7], requiring joint profiling of epigenetics and gene expression and their integration using multi-omics analyses.

Our current understanding of the neurobiological basis of CocUD implicates the involvement of a number of different brain regions involved in the development and maintenance of this disorder. Most notable are the ventral striatum, which is involved in reward processing as part of the mesocorticolimbic pathway[8], and the dorsal striatum—consisting of the putamen and caudate nucleus—which plays a critical role in habit formation and decision-making[9,10]. Most of our current knowledge comes from preclinical studies, with the majority investigating gene expression in the ventral striatum, particularly the nucleus accumbens. To date, only a few studies have focused on human postmortem brain tissue. An early study using microarray technology identified a reduction in myelin-related genes in the nucleus accumbens in CocUD[11]. Studies in other brain regions, such as the hippocampus, highlight the relevance of metabolic alterations, particularly in mitochondria, in CocUD[12], as well as changes in ion- and potassium-channel expression and receptor density[13]. While studies in the human striatum are sparse, more evidence is available from rodent models. On the bulk level, several studies highlight CREB-signaling[14,15], immediate early genes[14–16], and ion-channel expression[17–19]. Interestingly, a recent study showed that transcriptomic signatures associated with CocUD widely overlap in the nucleus accumbens and caudate nucleus in postmortem human brain tissue[19]. Furthermore, differential gene expression in human postmortem brain tissue was concordant with a mouse model, highlighting the translational value of such model systems in substance use disorders[19].

One major limitation of bulk-level studies results from the lack of cell type specificity of gene expression alterations. So far, cell type-specific gene expression studies have only been performed in the rat nucleus accumbens and drosophila models of CocUD. These studies suggested the importance of D1-medium spiny neurons (D1-MSNs)[20] and confirmed cell type-specific alterations in immediate early gene expression[21].

A second research gap depicts the lack of integrative multi-omics analyses in the same brain tissue specimens at single-cell resolution. Multi-omics analysis allows for a more mechanistic investigation of molecular alterations by combining evidence from multiple layers. With the 10x Genomics Multiome technology gene expression and open chromatin states can be profiled from the same cell, providing a powerful tool for linking epigenetic mechanisms to gene expression changes in CocUD. Here, we can directly infer transcription factor activity and construct gene regulatory networks (GRN) to identify upstream regulators of transcriptional changes.

In this work, we address these two research gaps by profiling the differential gene expression patterns and chromatin accessibility landscapes in a cell type-specific manner within the caudate nucleus of individuals with CocUD compared to non-affected subjects. By employing state-of-the-art sequencing techniques and data analysis, we demonstrate *ZEB1* as a key upstream regulator of transcriptional changes that underlie the pathophysiology of CocUD. We present druggable downstream targets with *PDE10A* and *CACNA1C* and validation experiments in the well-established 3-crit model of CocUD in rats[22,23]. Here, we profile human CocUD at the single-cell multi-omics level, in a highly affected but underrepresented population.

## Results

### Identification of cell types in the human caudate nucleus
We isolated nuclei from the human caudate nucleus of six individuals with CocUD (mean age 55) and eight controls (mean age 57), all of African American ancestry (Fig. 1A). CocUD cases and controls were balanced according to age and sex, see also Supplementary Data 1. We performed single-nuclei ATAC-seq and RNA-seq using the 10x Genomics Multiome platform, which allows for simultaneous profiling of chromatin accessibility and gene expression from each cell. After quality control, we retained 30,030 nuclei with a median of 1641 expressed genes and 14,418 ATAC fragments per nucleus. For normalization and dimension reduction, we used Seurat[24] and Signac[25]. The R package Harmony[26] was used for the integration and reduction of batch effects (Fig. 1B–D, Supplementary Fig. 1A–E). Weighted nearest neighbor analysis resulted in an integrated UMAP depicting 13 clearly delineated cell types, including astrocytes, microglia, oligodendrocytes, oligodendrocyte precursor cells (OPC), endothelial cells and lymphocytes, as well as different GABAergic neurons, such as D1 and D2 medium spiny neurons (MSNs) and a small population of cholinergic neurons (Fig. 1B). Distribution of cell types was comparable between samples, with one control sample having a higher ratio of astrocytes to oligodendrocytes (Fig. 1E). Annotation of clusters was supported by the expression of known marker genes (Fig. 1F), as well as their promoter accessibility (Fig. 1G) and cross-referenced with a recent publication profiling the dorsal striatum in opioid use disorder[27]. As expected in this brain region[28], the main neuronal cell types observed were GABAergic D1 and D2 medium spiny neurons (MSN).

### Differential expression and accessibility analysis
We performed within cell type differential expression analysis comparing CocUD cases with controls. Here, we identified 1485 differentially expressed (DE) genes (Supplementary Data 2), clustering distinctly according to glial or neuronal cell identity (Fig. 2A). Strongest deregulation of gene expression levels was observed in astrocytes (368 unique DE genes) and medium spiny neurons (D1: 20, D2: 80, D1/D2: 201 unique DE genes). The top-upregulated DE gene in astrocytes was *CTNNA3* (log2FC = 2.54, $p < 2.2e^{-16}$), belonging to the vinculin/alpha-catenin family, while the top downregulated gene was *LSAMP* (log2FC = −0.83, $p < 2.2e^{-16}$), which encodes the limbic system associated preprotein involved in axon guidance and neuronal growth. In both D1- and D2-MSNs, the top upregulated gene was the ferritin heavy chain 1 *FTH1* (log2FC$_{D1}$ = 2.09, $p_{D1} < 2.2e^{-16}$; log2FC$_{D2}$ = 2.32, $p_{D2} < 2.2e^{-16}$) and the top downregulated gene *SLC35F3*, a thiamine transporter (log2FC$_{D1}$ = −1.76, $p_{D1} < 2.2e^{-16}$; log2FC$_{D1}$ = −1.04, $p_{D1} < 2.2e^{-16}$). Interestingly, we observed an overall large overlap between DE genes in D1- and D2-MSNs (Fig. 2B). Gene Ontology (GO) enrichment analysis revealed an overrepresentation of DE genes from all clusters in ribosomal pathways, further a neuronal-specific cluster related to the respiratory chain and a cluster related to neuron projection evident in astrocytes, microglia, D1- and D2-MSNs, as well as a synaptic cluster in all cell types (Fig. 2D). In a subsequent KEGG analysis, we observed enrichment for addiction-related pathways, such as morphine addiction, and pathways related to neurodegenerative diseases (Fig. 2E).

Differential accessibility analysis revealed 10,342 differentially accessible (DA) peaks (Supplementary Data 3), again with a clear distinction between glial and neuronal cell types (Fig. 2F). The highest number of DA peaks was observed in astrocytes and oligodendrocytes (Fig. 2C). DA peaks were overrepresented in promoter and exonic regions, compared to the genomic background (Fig. 2G, H, Supplementary Data 4). In a GO term enrichment analysis performed separately for promoter and distal peaks, we observed an astrocyte-specific overrepresentation of ion- and cation channel activity promoter peaks (Supplementary Fig. 2A) and ligand- and voltage-gated channel and GTPase activity in distal peaks (Supplementary Fig. 2B).

### CocUD-specific peak-linked genes are associated with ribosomal activity and synaptic pathways
To identify CocUD-specific linked peaks, we linked peaks to gene expression separately for samples from individuals with and without CocUD, revealing N = 1050 CocUD-specific, N = 4578 common, and

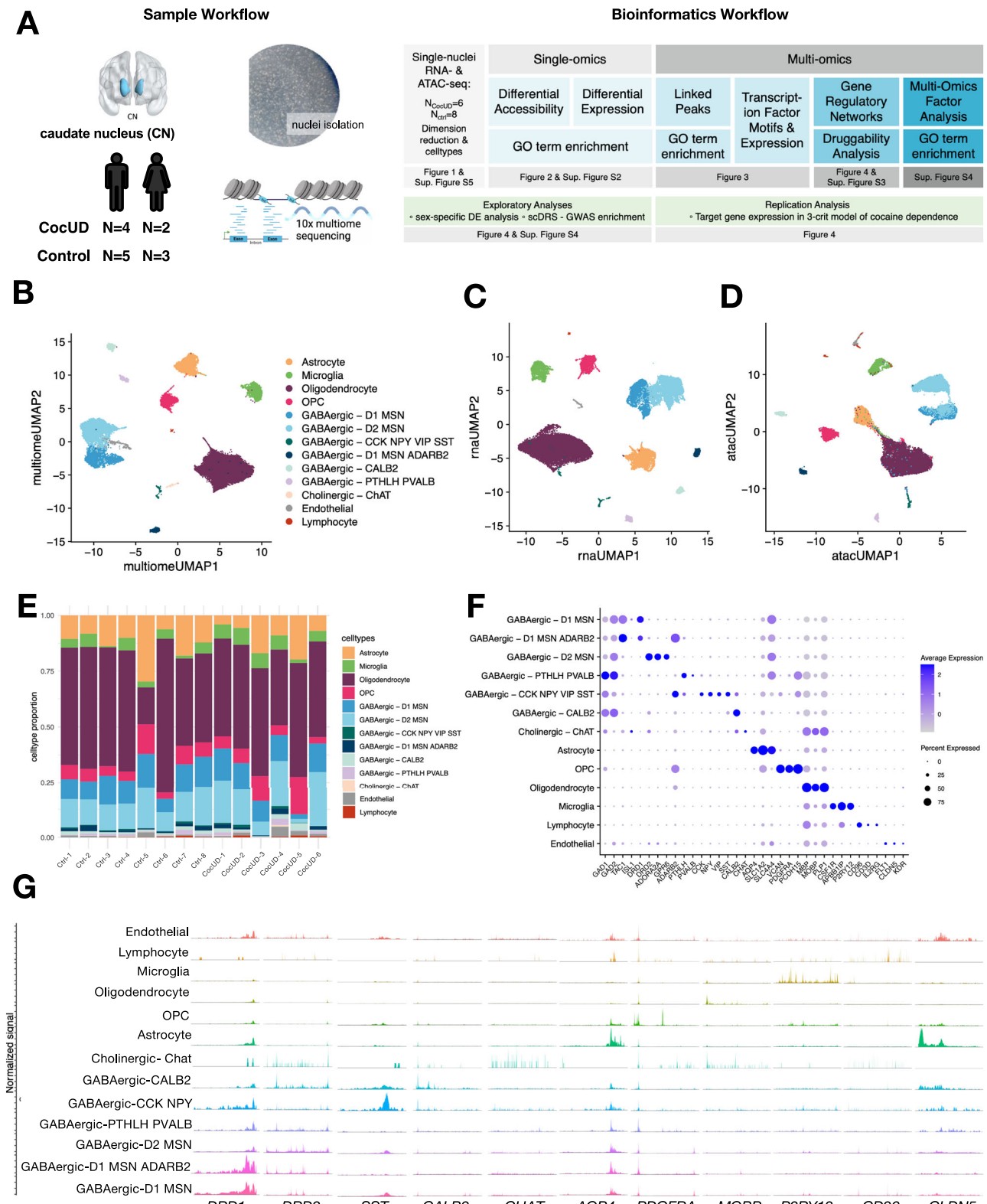

**Fig. 1 | Study workflow and characterization of single-nuclei multiome dataset.**
**A** Sample workflow for $N = 14$ individuals, top left image was created with BrainNet Viewer (v.1.7; Xia et al.[66], http://www.nitrc.org/projects/bnv/), and the bottom right image was created in BioRender. Espinola, M. (2025) https://BioRender.com/c45a312; and bioinformatics workflow relating each analysis to the corresponding figure. **B** Integrated UMAP for 30,030 nuclei, depicting 13 clearly delineated cell types. **C** UMAP of the RNA assay. **D** UMAP of the ATAC assay. **E** Cell type proportions per sample. **F** Dot plot depicting marker gene expression per cell type. **G** Coverage plot depicting chromatin peaks in promoter regions of cell type marker genes. CN = caudate nucleus, CocUD = cocaine use disorder, UMAP = uniform manifold approximation and projection.

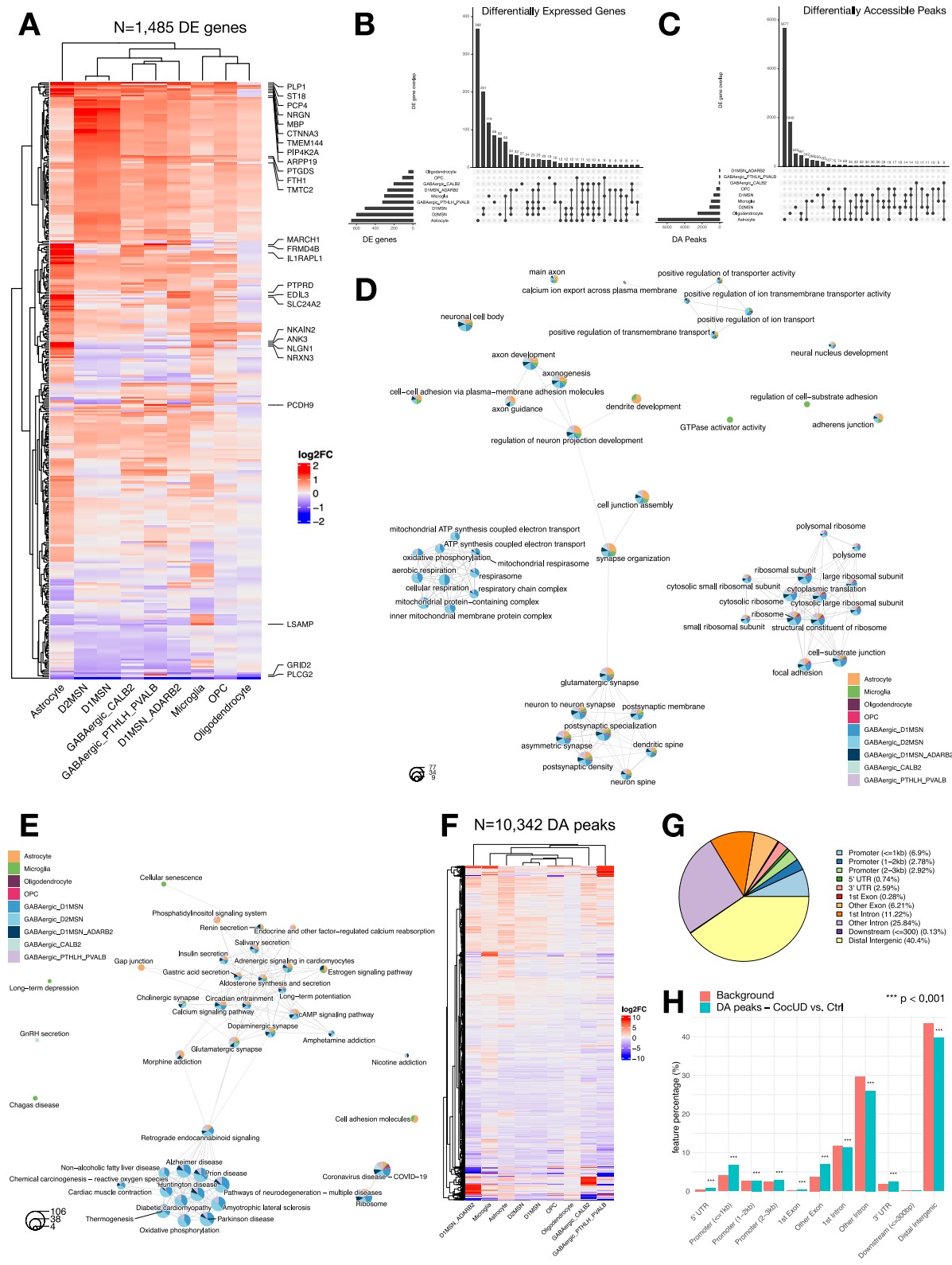

*N* = 1609 control-specific gene-peak pairs (Fig. 3A). As multiple peaks can be linked to a single gene, we then further investigated the functional relevance of linked genes. GO term enrichment analysis of CocUD-specific peak-linked genes showed overrepresentation of ribosomal and synaptic pathways (Fig. 3B) thereby supporting results from cell type-specific differential expression analyses. We thus concluded that altered chromatin accessibility patterns might be an

important factor contributing to the observed expression changes of ribosomal and synaptic genes in CocUD.

## Family-wise motif enrichment of transcription factors in cell types

Altered open chromatin states in CocUD can increase or decrease the availability of transcription factor motifs thereby resulting in

**Fig. 2 | Differential expression and accessibility analysis of cocaine use disorder and GO overrepresentation. A** Heatmap depicting differentially expressed genes identified using a two-sided Wilcoxon rank sum test (Bonferroni-adjusted $p < 0.05$, log2FC > 0.5, expressed in at least 25% of cells), red = upregulated in CocUD, blue = downregulated in CocUD, top ten DE genes per cell type are labeled. Upset plot depicting the number of shared and distinct differentially expressed features per cell type, for **B** differential expression and **C** differential accessibility. **D** emapplot depicting the top 14 category results of the Gene Ontology overrepresentation analysis for DE genes in a cell type-specific manner. Each circle represents one GO term as a pie chart, indicating the strength of enrichment in the different cell types, represented by color. Edges indicate semantically similar GO terms, with overlapping genes. Circle size indicates the number of genes in the pathway. Overrepresentation was determined using a Fisher's exact test and Benjamini-Hochberg adjustment for multiple testing. **E** emapplot depicting the top 14 category results of the Kyoto Encyclopedia of Genes and Genomes (KEGG) overrepresentation analysis for DE genes in a cell type-specific manner. Each circle represents one GO term as a pie chart, indicating the strength of enrichment in the different cell types, represented by color. Edges indicate semantically similar GO terms, with overlapping genes. Circle size indicates the number of genes in the pathway. **F** heatmap showing differentially accessible (DA) peaks identified using a two-sided Wilcoxon rank sum test (Bonferroni adjusted $p < 0.05$, log2FC > 0.25, expressed in at least 5% of cells), red = upregulated in CocUD, blue = downregulated in CocUD. **G** Annotation of DA peaks to their genomic location. **H** comparison between genomic location of DA peaks to the genomic background, using a two-sided chi-square test with Bonferroni adjustment for multiple testing, *** = $p < 0.001$, ** = $p < 0.01$, * = $p < 0.05$.

---

transcription factor-mediated downstream effects on gene expression. To evaluate differential chromatin accessibility related to transcription factor motifs, we performed a cell type-specific motif enrichment analysis (Fig. 3C) confirming cell type-specific transcription factor motif activity patterns in cell types of the caudate nucleus (Fig. 3D). Next, we investigated differential motif activity across cell types in a CocUD vs. Ctrl comparison. We observed major downregulation of transcription factor (TF) motif activity in CocUD of immediate early genes, such as *JUND* and *FOS*, in D1- and D2-MSNs, while the same motifs were upregulated in astrocytes, oligodendrocytes and OPCs (Fig. 3E and Supplementary Data 5). Strong effects were also present for the SOX and RFX transcription factor families exhibiting prominent activation and deactivation patterns in CocUD, respectively (Fig. 3E). To determine whether there was differential TF expression, alongside differential motif activity, we specifically investigated differential expression of TFs previously identified in the motif enrichment analysis (Fig. 3F). Here, the strongest differential expression patterns were observed for *ZEB1, RFX3, FOXP2,* and *NFIB*. For *ZEB1*, we observed reduced availability in CocUD of its motif (Fig. 3G) in neurons while increased availability in astrocytes was found. At the same time, statistically significant downregulation of the *ZEB1* transcript was found across neuronal and astrocyte clusters in CocUD. We confirmed robust expression of *ZEB1* in neuronal and astrocyte clusters across conditions (Fig. 3H) implicating it is a promising candidate transcription factor for inducing downstream differential expression in these cell types. Another deregulated transcription factor in CocUD was *FOXP2*, for which we found altered motif availability and transcript expression especially in D1-MSNs and microglia (Fig. 3G, H).

## Gene regulatory networks reveal TFs accounting for transcriptional changes in CocUD

Following up on the observation of common transcriptional changes in D1- and D2-MSNs and the identification of transcription factor candidates such as *ZEB1* and *FOXP2*, we were interested in GRN related to CocUD. GRNs can be used to infer the relationship between TFs and their downstream targets (Fig. 4A). An MSN-specific GRN was constructed using pando[29] and metrics are depicted in Supplementary Data 6 and Supplementary Fig. 3A. As immediate early genes are known to influence gene expression of a large set of downstream targets after exposure to multiple drugs of abuse, we were interested if there are conserved sets of TFs and their targets in our CocUD dataset that might be of particular importance for the induction of altered transcriptional programs. Appropriately, two subclusters emerged in the GRN, one consisting mainly of immediate early genes, such as *FOSB* or *JUND*, targeting heat shock proteins (e.g., *HSPA1B, HSPA1B*) and downstream map kinases (Fig. 4A). A larger cluster encompasses the TFs highlighted in the motif analysis such as *ZEB1* and *FOXP2*, targeting calcium-, potassium- and ion-channels. To identify the most relevant TFs, we performed a scoring approach wherein the size of the TF regulon is weighed by the log2FC from a differential regulon expression analysis (Supplementary Fig. 3B & Supplementary Data 7). The regulon refers to all downstream targets that are either activated or suppressed by the associated TF based on the GRN prediction in Pando (Supplementary Fig. 3C). Hence, one TF can have two regulons for activated downstream genes, e.g., ZEB1(+), and for suppressed ones, e.g., ZEB1(-). By investigating the regulons, we observed that a small number of TFs account for a large number of downstream target genes, many of which have been previously identified in the DE analysis (Supplementary Fig. 3D). A TF with particularly large effects in both activating and suppressing downstream gene expression was again *ZEB1* that is widely expressed in D1- and D2-MSNs (Supplementary Fig. 3B, C; Supplementary Data 7). Interestingly, we observed that *ZEB1* has specific suppressing effects in controls and activating effects in individuals with CocUD: CocUD-specific subclusters of MSNs express downstream targets activated by *ZEB1*, while suppressed *ZEB1* targets are preferably expressed in control individuals (Fig. 4B). GO overrepresentation analysis of the downstream targets of the three most important TFs derived from the GRN, *ZEB1, PBX3*, and *ZNF148*, revealed common suppressing effects on genes related to ribosomal pathways while activated target genes were consistently involved in synaptic signaling pathways (Fig. 4C). To further explore the GRN in MSNs, we characterized the chromatin structure of highly connected TF targets, such as *PDE10A* and *CACNA1C* by visualizing differential accessibility in TF binding sites between individuals with CocUD and controls (Supplementary Fig. 3E & F).

## Druggability analysis suggests phosphodiesterase-inhibitors as potential pharmacological targets

When identifying potential pharmacological targets for disorders with complex molecular alterations, it is crucial to preserve this complexity in druggability analysis. Therefore, we performed a druggability analysis of highly connected target genes from the D1-/D2-MSN GRN, using the drug-gene interaction database (DGIdb). Here, we observed several pharmacological agents with the potential to treat CocUD (Supplementary Data 8). Interestingly, among the top drug targets with large effect size in the DE analysis, we observed PDEs (*PDE10A* and *PDE7B*) several times, suggesting brain-expressed phosphodiesterases as a potential treatment target in CocUD.

## Validation of differential expression in *PDE10A* and *CACNA1C* in the 3-crit model of CocUD

We prioritized the DE genes from the CocUD DE analyses based on their relationship with the DA peaks (*CACNA1C*), their involvement in the synaptic signaling GO terms (*KCNIP4*; Supplementary Data 9), their role in the GRN (*ZEB1*) and the druggability analysis (*PDE10A*). To validate these findings, we performed RT-qPCR analysis on mRNA extracted from the dorsal striatum of rats that underwent the 3-crit model of CocUD, a cocaine self-administration procedure after which three addiction-like criteria are assessed: persistence of drug seeking, motivation for drug taking, and resistance to punishment. Animals fulfilling all three criteria are classified as "addicted-like". We compared five 3-crit animals to six 0-crit animals that were exposed to cocaine

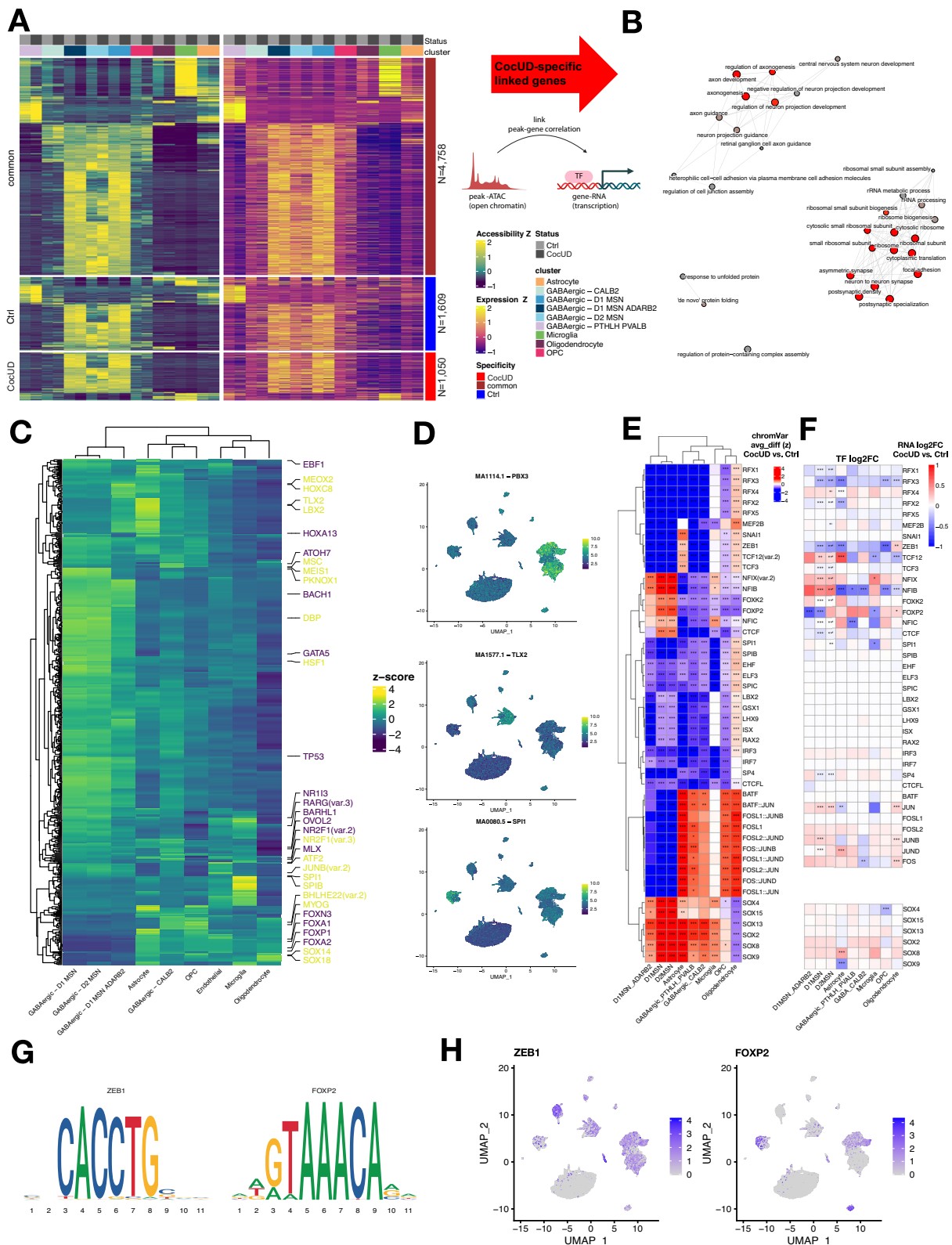

but did not develop addiction-like symptoms and six cocaine-naïve control animals ($N_{total} = 17$, Fig. 4D). We observed differential expression in *PDE10A* ($p = 0.015$), consistent with an upregulation in the non-neuronal cell types (Supplementary Data 2), and in *CACNA1C* ($p = 0.026$), in line with an upregulation in microglia and oligodendrocytes, while a downregulation was observed in MSNs in the DE analysis (Supplementary Data 2). These effects were significant for 0-crit, but not 3-crit animals, compared to controls, indicating either a

cocaine-consumption effect or a habituation in the 3-crit animals. No effect was observed for *ZEB1* ($p = 0.82$) and *KCNIP4* ($p = 0.82$).

**Multi-Omics factor analysis reveals a sex-specific factor characterized by metabolic pathway enrichment**

In addition to the data-informed approaches, we performed a data-driven multi-omics factor analysis to identify a dimensionality-reduced set of factors associated with CocUD. MOFA factors capture both RNA

**Fig. 3 | Linked peaks and transcription factor motif and expression analysis.**
**A** Heatmap depicting shared (brown), CocUD-specific (red), and control-specific (blue) links between chromatin accessibility and RNA expression by cell type.
**B** Emapplot showing overrepresented gene ontology biological processes for genes characterized by a CocUD-specific linked peak in (**A**) color represents statistical significance and circle size the number of genes in the GO term; overrepresentation was determined using a Fisher's exact test and Benjamini-Hochberg adjustment for multiple testing. **C** Heatmap depicting cell type-specific motif enrichment, yellow motif names indicate increased activity while motif names in blue represent reduced activity of the motif in cell types based on chromVAR

activity scoring. **D** Feature plots confirming motif expression for MA1114.1/*PBX3* in D1- and D2-MSNs, MA1577.1/*TLX2* in astrocytes and MA0080.5/*SPI1* in Microglia. Enrichment scores are color-coded from blue (no enrichment) to yellow (strong enrichment). **E** Heatmap depicting differential transcription factor motif activity between CocUD and controls, red = increased activity in CocUD, blue = decreased activity in CocUD. **F** Heatmap depicting corresponding transcription factor differential transcript expression in CocUD, red = upregulated in CocUD, blue = downregulated in CocUD. **G** Transcription factor motifs for *ZEB1* and *FOXP2*, size of sequence represents probability. **H** Feature plots depicting *ZEB1* and *FOXP2* expression in cell types of the caudate nucleus (log-normalized counts).

and ATAC information and are therefore particularly informative about multi-modal (i.e., multi-omics) interactions between RNA and ATAC signals. Of the 15 resulting factors, four were associated with CocUD (Supplementary Fig. 4A, B). Factor5 is characterized by high factor loadings for GABAergic CALB2 and PTHLH PVALB cells, with positive enrichment for synaptic signaling and neurodevelopment and negative enrichment for pathways related to synapse maturation and gliogenesis (Supplementary Fig. 4B, C).

Factor7 showed a CocUD-specific association with D2-MSNs and was enriched for GO terms related to the ribosome and translation pathways (Supplementary Fig. 4D). Overall, the results for the CocUD-associated factors confirmed the findings of the DE analysis.

Of particular interest is the CocUD-associated Factor8 that indicated sex-specific subclusters in the majority of cell types, marked by high factor-loadings in subclusters of OPCs, oligodendrocytes, and particularly D1- and D2-MSNs. Characterized by an enrichment for metabolic GO terms, this factor suggests a male-specific effect. Interestingly, in the promoter-associated motifs of Factor8, we observe a negative enrichment of major regulators from the GRN, such as *ZEB1* and *TCF4* (Supplementary Fig. 4E), also in line with the DE and chromVAR analyses.

### Sex-stratified DE analysis shows concordant effects in astrocytes, microglia, and D1-/D2- MSNs
Because we observed high Factor8 loadings in the majority of cell types, we performed exploratory analyses on potential sex-specific transcriptional signatures of CocUD. Using rank-rank hypergeometric overlap (RRHO) analysis, we aimed to identify concordant and discordant CocUD-associated transcriptional changes in individual cell types. Interestingly, we observed high concordance of CocUD-associated DE patterns between sexes in astrocytes, D1-MSNs, and microglia, and moderate concordance between sexes in D2-MSNs, and OPCs (Fig. 4E). Strong divergent patterns were observed in oligodendrocytes suggesting sex-specific transcriptional changes in these clusters which further underscores the results from MOFA.

### No cell type-specific enrichment of GWAS signals of CocUD
To test whether GWAS signals of substance use disorders and CocUD in particular, were overrepresented in cell type gene expression patterns, we used single-cell disease-relevance scoring (scDRS). In scDRS, a polygenic enrichment analysis is performed for each cell to evaluate whether its expression profile is significantly enriched for phenotype-associated risk genes. In microglia, oligodendrocytes, and cholinergic neuron clusters, we found the strongest CocUD-associated enrichment patterns in individual cells similar to the results observed for alcohol dependence (Supplementary Fig. 4F). However, we did not observe statistically significant enrichment for any of the investigated SUDs, either at the single cell or the cluster level, most likely due to the small GWAS sample sizes in AA populations (Supplementary Data 10, Methods section).

## Discussion
Our single-nuclei multi-omics study of CocUD in the human caudate nucleus reveals insights into cell type-specific alterations of gene

expression and the underlying chromatin states, directly linking transcription factor activity to downstream gene expression. Here, we investigate human CocUD at single-nuclei resolution combining ATAC- and RNA-seq in a multi-omics approach in the same cell. Our study highlights synaptic pathways, ribosomal activity, and metabolic pathways as altered in CocUD, and demonstrates that these processes are deregulated in a cell type-specific manner.

We present different lines of analysis from data-informed differential gene expression and chromatin accessibility analyses to data-driven GRN and factor analyses. We prioritized results that showed convergent evidence from the different lines of analysis.

In both the DE and DA analyses, we observed initial evidence for altered ion- and calcium-channel activity, implicating altered synaptic signaling in CocUD. In addition, we found an overrepresentation of genes in pathways related to synapse organization in CocUD. Changes in synaptic plasticity following repeated cocaine exposure have been well described[30,31]. This is also consistent with the large effects observed for ribosomal genes, both in the DE, DA and GO overrepresentation analyses, as the transcription of genes encoding the protein components of ribosomal subunits is an important prerequisite to neural plasticity[32,33]. Interestingly, we also observed KEGG-enrichment of neurodegenerative diseases and addiction-related processes in the neuronal cell types, providing evidence that our results reflect known pathologies. In addition, we observed vast expression differences in astrocytes related to synaptic processes. Consistent with our results, it has been shown that cocaine can trigger astrocyte-mediated synaptogenesis in the nucleus accumbens shell[34].

To exploit the strength of the multiome framework, the simultaneous profiling of RNA and open chromatin in the same cell, we calculated a GRN in D1- and D2-MSNs. Here, we observed two clusters, the smaller one consisting of immediate early genes, whose effect has been described extensively[21,35]. The other cluster consisted of a set of major regulators, with downstream effects mainly on calcium-, ion-, and potassium-channels that have also been found in the differential expression analysis. Interestingly, we were able to show CocUD-specific effects, not on TF activity per se, but on the activation and suppression of downstream gene expression. We observed that *ZEB1* activates genes related to synaptic signaling in a CocUD-specific subcluster of MSNs, *ZEB1* could therefore potentially mediate this well-described effect[30]. Interestingly, *ZEB1* was also recently identified as a major transcriptional regulator in Alzheimer's disease[36], providing another link between neurodegenerative diseases and SUDs. We further explored the druggability of the highly connected downstream targets, revealing phosphodiesterase inhibitors as a candidate drug, consistent with previous studies in rodent models showing reduced acquisition of cocaine place preference and cue-induced reinstatement of cocaine seeking[37,38]. Because of the large overlap of DE genes in the D1- and D2-MSNs, we constructed the GRN using information from both cell types. It has to be noted that while the transcriptional signatures of the two cell types are largely overlapping, a recent study has shown that transcriptional changes in D1-MSNs are more prevalent in the acute response to cocaine and in D2-MSNs more reflective of a cumulative effect of chronic exposure[39].

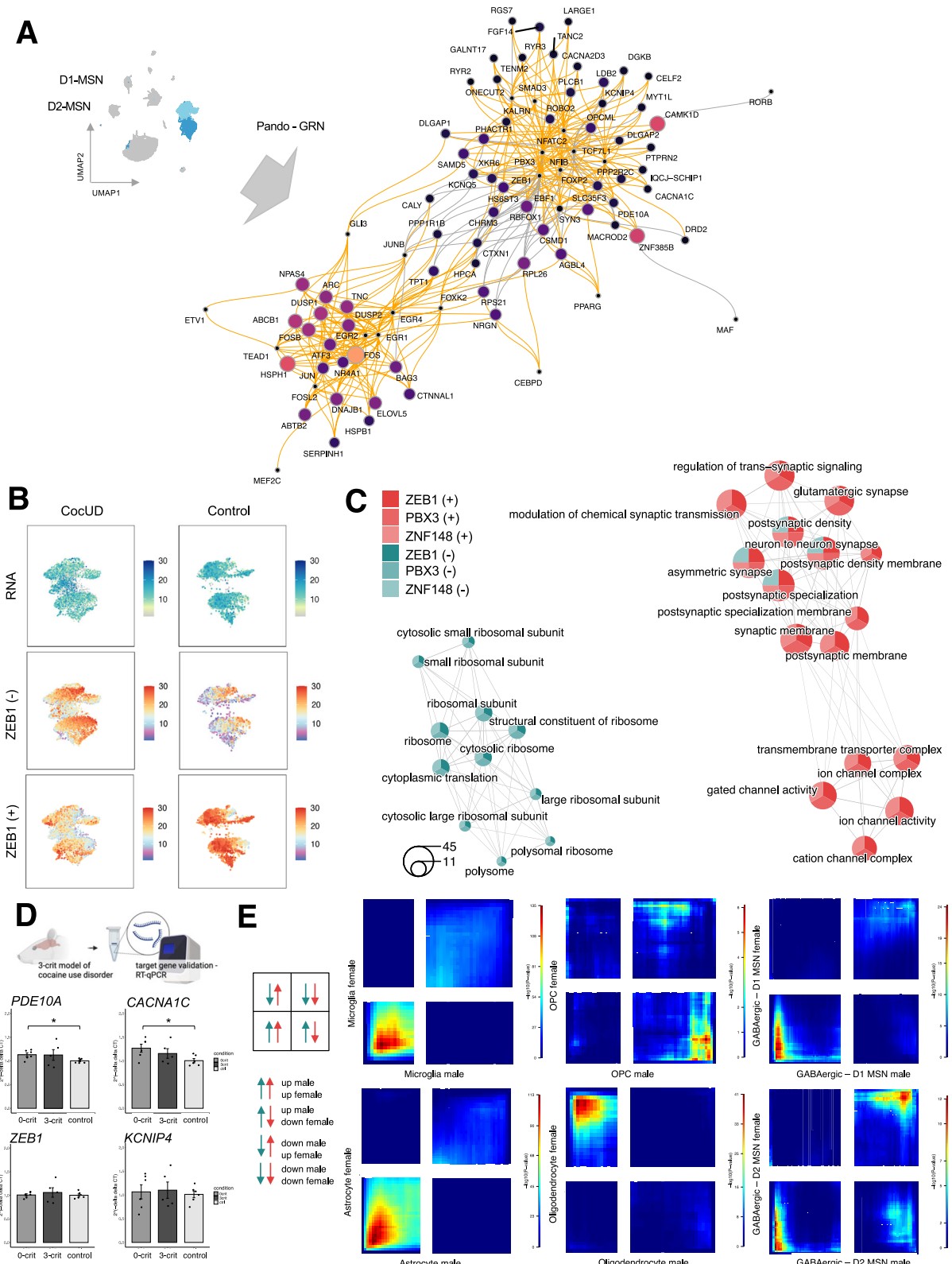

A highly connected downstream target in the GRN with large effects in the DE analysis was *CACNA1C*, calcium voltage-gated channel subunit alpha1 C, which has been broadly implicated in psychiatric disorders[40]. Further, there is evidence for an important role of *CACNA1C* in cocaine addiction. In a rodent model of CocUD it has been shown that *CACNA1C* heterozygous mice demonstrate an increased reward response to cocaine[41]. In another study, L-type

calcium channel blockage by nifedipine was associated with less neurotoxicity in the prefrontal cortex of rats exposed to acute cocaine[42]. At the same time, we observed a downregulation of *CACNA1C* in the DE analysis in MSNs and an upregulation in cocaine-exposed animals in the 3-crit model, emphasizing the importance of distinguishing acute and long-term exposure and cell type-specific effects.

**Fig. 4 | Gene regulatory network of D1- and D2-MSNs, candidate gene validation in the 3-CRIT model and sex-specific RRHO analysis. A** Gene Regulatory Network (GRN) from the D1- and D2-MSNs, color and size of nodes relate to centrality in the network, larger circles represent increased centrality, orange edges represent activation, and gray edges inhibition. **B** Feature plots depicting *ZEB1* RNA expression (upper), module scores for *ZEB1* suppressed downstream targets (middle), and module scores for *ZEB1* activating downstream targets (lower), separately for CocUD (left) and controls (right). **C** Emapplot summarizing GO overrepresentation results from genes activated (red) and suppressed (turquoise) by ZEB1, PBX3, and ZNF148, the top TFs from regulon scoring. Each circle represents one GO term as a pie chart, indicating the strength of enrichment in the different regulons, represented by color. Edges indicate semantically similar GO terms, with overlapping genes. Circle size indicates the number of genes in the pathway. **D** Summary of validation analysis in 3-crit model of cocaine use disorder in rats, upper panel was Created in BioRender. Espinola, M. (2025) https://BioRender.com/c99g053, data are presented as mean values +/- SEM, light gray = controls ($N = 6$), gray = 0crit ($N = 6$), black = 3crit ($N = 5$), differential expression was determined using a Wilcoxon rank sum test, $p_{PDE10A} = 0.015$, $p_{CACNA1C} = 0.026$, $p_{KCNIP4} = 0.82$, $p_{ZEB1} = 0.82$. **E** RRHO plots showing the concordance between DE results in males and females, rank rank hypergeometric overlap tests were used to identify sex-specific concordance, upper left = upregulated in females and downregulated in males, upper right = upregulated in males and females, lower left = downregulated in males and females, lower right = upregulated in males and downregulated in females. MSNs = medium spiny neurons, GRN = gene regulatory network, CocUD = cocaine use disorder, Ctrl = control, (+) = activating, (-) = suppressing.

We explored whether broad gene expression changes of GRN-identified targets were observed in the 3-crit model of CocUD reflecting addiction-like criteria. Here, we validated the effects for *PDE10A* and *CACNA1C* in 0-crit, but not 3-crit animals relative to naïve controls. This finding could suggest that alterations in *PDE10A* and *CACNA1C* may be due to long-term cocaine exposure rather than CocUD, per se, and highlights an important challenge in addiction research: based on postmortem human brain tissue alone, it is difficult to conclude whether the observed effects result from substance intake or reflect neuroadaptations resulting from substance use disorder. However, because 0- and 3-crit animals don't differ in cocaine exposure, it is likely that differences between these two groups in *PDE10A* and *CACNA1C* relative to controls reflect neuroplastic/neuroadaptive changes in response to an addiction-like phenotype not seen in the human samples. Furthermore, the 3-CRIT model assesses three specific criteria of CocUD in humans that are modeled for rats[22], such that this model does not perfectly represent the diverse symptomology of CocUD in humans. Nonetheless, our study demonstrates some convergence among CocUD in humans and a rat model, and represents a unique strategy for potential validation of targets identified by multi-omics from human tissue.

In addition to the GRN, we performed another data-driven analysis, Multi-Omics Factor Analysis (MOFA). MOFA confirmed the results of the DE analysis, with the neuronal-specific factor pointing towards increased metabolic processes and synaptic signaling in CocUD. We also observed a microglia-specific factor associated with control status, providing further evidence that neuroinflammation is not elevated in CocUD[19], as opposed to other SUDs, such as alcohol use disorder[43,44] and opioid use disorder[19]. Interestingly, we also observed a sex-specific factor associated with CocUD and enriched for ribosomal and metabolic processes. While sex-specific effects have been observed at large in rodent models, suggesting that female rats are more susceptible to cocaine consumption, evidence from human studies suggests an effect specific to the luteal phase of the menstrual cycle[45]. In our exploratory analyses of potential sex-effects in the gene expression data, we observed strong sex-specific effects in oligodendrocytes, endothelial cells, and CALB2-GABAergic neurons, providing evidence into potential drivers of sex-specific effects in CocUD.

At the same time, the sex-specific effects need to be interpreted with caution. While the single-nuclei framework allowed for a high statistical power, because each cell was treated as one observation, the overall sample size was still small and results could lack generalizability. Larger studies are needed to conclusively investigate sex-specific effects in CocUD. Similarly, the overall sample size in the current study and the number of nuclei per cell type remain a limitation that requires more data generation and integration. Larger sample sizes would also allow for using state-of-the-art statistical methods, such as hierarchical modeling, less susceptible to false positives than the Wilcoxon rank sum test, while maintaining the single-cell resolution.

We did not observe cell type-specific enrichment of GWAS signals of CocUD. It has to be noted that the currently available GWAS samples of CocUD are small, comprising 2482 cases and 836 controls of African American ancestry, and have not yet identified genome-wide significant SNPs[46]. Future studies increasing the sample size of CocUD GWAS are needed[5], providing the basis to interpret GWAS signals in multi-omics studies.

Our results replicate known neurobiological mechanisms in CocUD and identify targets, with complex CocUD interaction effects, particularly *ZEB1*, for future investigation. The present analysis highlights the potential of using a multi-omics framework allowed us to analyze the data in a more mechanistic approach compared to single-omics datasets.

## Methods
### Human tissue
**Postmortem human brain tissue.** Postmortem human brain tissue samples of the caudate nucleus from six individuals with and eight individuals without CocUD of African American descent were obtained from the University of Texas Health Science Center at Houston (UTHealth Houston) Brain Collection (Supplementary Data 1). CocUD donors had age >18 ($M = 54.17$, SD = 9.02) and a lifetime diagnosis of cocaine use disorder based on next-of-kin interviews and psychological autopsy[47]. Controls had no history of substance use or neuropsychiatric disorders (mean age: $M = 57.13$, SD = 14.37). For all cases, exclusion criteria included a known diagnosis of severe neurodevelopmental disorders. Postmortem tissue was sampled based on written informed consent from each donor's next of kin. The study was conducted in line with the guidelines of the Declaration of Helsinki. It was approved by the Ethics Committee II of the Medical Faculty Mannheim under the register number 2021-681 and by the University of Texas Health Science Center at Houston (HSC-MS-15-0247).

**Nuclei isolation.** For each sample, nuclei were isolated from 10 mg of frozen postmortem tissue, in order to minimize the content of free-floating debris. Brain samples were processed following the 10x Genomics Chromium Nuclei Isolation kit with RNase Inhibitor user guide. Briefly, the tissues were dissociated in lysis buffer, filtered and the cellular debris eliminated with the designated buffer. Then, alternate steps of centrifugation and wash resulted in clean nuclei suspensions. Nuclei were automatically counted on LUNA-FL™ Dual Fluorescence Cell Counter and particles smaller than 5 μm of diameter and bigger than 15 μm of diameter were excluded from the counting.

**Library preparation and sequencing.** Single-nuclei RNA-seq and ATAC-seq libraries were generated using the 10x Genomics Chromium Next GEM Single Cell Multiome ATAC + Gene Expression kit and user guide. In brief, single nuclei were loaded into chromium chips J with a capture target of 10,000 nuclei per sample. Final libraries were prepared as indicated in the 10x Genomics protocol and sequenced at

NGS Competence Center Tübingen (Tübingen, Germany) using an Illumina NovaSeq6000 sequencer with a depth of 50,000 read pairs per nuclei for the RNA libraries and 25,000 read pairs per nuclei for the ATAC libraries.

## Single nuclei RNA- and ATAC sequencing data analysis

**Preprocessing and quality control.** Raw data obtained from single-nuclei RNA- and ATAC-sequencing was preprocessed using the Cell Ranger ARC software (10x Genomics, v.2.0.2) using the human genome built GRCh38 as reference. Resulting filtered feature barcode matrices were imported in R (v.4.2.1) and converted to a Seurat object using Seurat v.4.3.0[24]. In Seurat, quality control was performed and nuclei were excluded if they did not meet the following parameters: number of RNA features between 200 and 6500, number of ATAC features between 1000 and 25,000, percentage of mitochondrial genes <10%, TSS enrichment score between 2 and 8, and nucleosome signal between 0.5 and 2. We used DoubletFinder[48] to identify doublets, two cells that were captured in the same GEM, and removed the doublets for each sample (Supplementary Data 1).

**Data integration and clustering.** After quality control, individual datasets from the 14 individuals were merged to a single Seurat object using the merge function. ATAC peaks were called in the merged object using MACS2[49] to obtain a uniform set of peaks. After normalization, variable feature identification, and data scaling of the RNA assay, we applied dimensionality reduction using the RunPCA function and integrated the expression values across the 14 individuals with harmony[26] (v.0.1.1) using the first 20 principal components from PCA, followed by UMAP reduction. In a similar approach using Signac (v.1.9.0)[25], the ATAC assay was normalized using TFIDF, and linear dimensionality reduction was performed using SVD. Based on the principal components 2-30, UMAP reduction was performed followed by data integration using harmony. Finally, the RNA and ATAC assays were combined using a weighted nearest neighbor methodology as implemented in the FindMultiModalNeighbors function in Seurat. Using a resolution threshold of 0.25, a total of 15 clusters were identified in the single-nuclei multiomic dataset.

**Identification of cell types.** The expression of known marker genes for cell types in the human striatum[50] and different types of expected interneurons was evaluated in the RNA assay. In addition, we utilized the FindAllMarkers function to generate differential expression profiles distinguishing expression in one cluster from all other clusters (Supplementary Fig. 5). We cross-checked highly expressed genes in clusters with the Transcriptomics Explorer of the Allen Human Brain Atlas[51] and the Human Protein Atlas[52] leading to the identification of 14 different cell type clusters. One minor cluster was characterized by strong expression of mitochondrial genes as evaluated by the FindAllMarkers function in R. We excluded this cluster expressing oligodendrocyte markers from the object and performed further analysis in the 13-cluster object. In addition, we compared our marker expression profile to the human dorsal striatum dataset from Phan et al.[27] and observed similarities between our D1-MSN-ADARB2 cluster and the D1/D2-MSN hybrid cluster identified in Phan, et al.[27]. At the same time, we only observed spurious DRD2 expression in our D1-MSN-ADARB2 cluster suggesting them as a D1-MSN subpopulation.

**Differential expression analysis.** Differentially expressed (DE) genes were determined based on a Wilcoxon rank sum test as implemented in the FindMarkers function in Seurat. As differential expression cut-offs, absolute log2-fold change >0.5, minimal expression in at least 25% of cells in either of the compared clusters and an adjusted $p < 0.05$ were used. DE analyses were restricted to clusters consisting of at least 100 nuclei in each condition (CocUD yes/no) to ensure sufficient statistical power in the comparison. Visualization of DE genes was performed using the R packages ComplexHeatmap[53] (v.2.14.0) and UpSetR[54] (v.1.4.0).

**Differential chromatin accessibility analysis.** Identification of differentially accessible (DA) peaks in the ATAC assay for each cluster was performed using FindMarkers based on peaks that were at least expressed in 5% of cells in each of the compared clusters. Differential accessibility cut-offs were absolute log2 fold-change >0.25 and adjusted $p < 0.05$. DA analyses also were restricted to clusters with at least 100 nuclei per condition.

**Genomic annotation and overrepresentation analysis of ATAC peaks.** Annotation of DA peaks to genomic features based on chromosomal coordinates of the GRCh38 reference genome was performed using the annotatePeak function of the R package ChIPseeker[55] (v.1.34.1). We further tested the enrichment of DA peaks in different genomic feature categories using the annotation of all peaks in the ATAC assay as the reference. A chi-squared test was used to test for significant enrichment within a category and Bonferroni-correction was used to adjust resulting $p$-values for multiple testing.

**Gene ontology enrichment analysis.** To test the overrepresentation of upregulated and downregulated DE genes in GO terms for the different cell type clusters, we used the compareCluster function of the R package clusterProfiler[56] (v.4.6.2) with the list of significant DE genes as input. Results for GO terms with a significant enrichment (adjusted $p < 0.05$) were visualized as pie plots in GO term networks using enrichplot (v.1.18.3). The same approach was applied on gene lists resulting from the annotation of CocUD-associated differentially accessible distal and promoter peaks to genes with the intention of identifying GO terms related to differential accessibility.

**Linking peaks to genes.** Peak-gene associations were identified using the LinkPeaks functionality of Signac. Here, the correlation between accessibility of peaks within a threshold window of max 500 kb distance from transcription start site and the expression level of respective genes is determined. To identify peak-gene correlations specific to either CoCUD or Ctrl, we performed the linking of peaks to genes individually in nuclei from CocUD and Ctrl individuals. We then performed an overlap analysis of the identified links, filtered for absolute correlation coefficients >0.1, and grouped the remaining links into three categories: CocUD-specific: peak-gene associations only identified in CoCUD, Ctrl-specific: peak-gene associations only identified in Ctrl and common: peak-gene associations identified in both, CocUD and Ctrl.

**Motif enrichment analysis.** To assess the enrichment of transcriptional motifs within DA peaks associated with CocUD status, we performed motif enrichment analysis (MEA) using chromVAR[57] (v.1.20.2). The JASPAR2020 (http://jaspar.genereg.net/) position frequency matrix was used to obtain a comprehensive and experimentally validated set of non-redundant human transcription factor motifs. First, the RunChromVar function was applied to the full ATAC assay to identify motif activities (z-scores) in the dataset. Up- and downregulated differentially active motifs that are associated with CocUD status were assessed for each cell type cluster using the FindMarkers function resulting in an estimate for the average difference in ChromVAR z-scores. Differentially active motifs with an adjusted $p < 0.05$ were considered statistically significant.

**Identification of gene regulatory networks.** We used pando[29] to infer GRN based on snRNA and snATAC data for each cluster using the respective DE genes as the candidate gene list for inferring GRNs. Pando uses multi-omic data together with inferred transcription factor binding sites to generate a GRN consisting of transcription factors and

their inferred targets. After GRN initiation and motif scanning based on non-redundant human JASPAR2020 TF binding motifs, transcription factor modules were identified using the following parameters: $p$-value threshold = 0.05, number of variables = 5, minimum number of genes per module = 5, and $R^2 = 0.1$. Network graphs for TFs and targets in each of the evaluated clusters were generated using the force-directed Fruchterman-Reingold layout.

**Druggability analysis of GRN targets.** Reference data from the drug-gene interaction database[58] (DGIdb) was used to evaluate drug-gene interactions and potential druggability of GRN targets. Gene targets from the GRN were prioritized based on the number of statistically significant TF-target associations inferred by Pando. We determined quantiles of the TF-target connectivity distribution and selected targets with a minimum number of $N = 17$ significant TF-target connections corresponding to the value related to the 95% quantile. All of the identified target genes were connected to multiple TFs (mean = 16 upstream TFs) in the GRN network highlighting their potential use as convergent downstream drug targets. The prioritized set of target genes was queried using the DGIdb online tool (https://www.dgidb.org, v.5.0.5) with "Interactions by Gene" mode. Results for unique matches of gene-drug interactions were downloaded and sorted by gene-drug interaction score and FDA approval state of connected drugs.

**Multi omics factor analysis.** We applied multi omics factor analysis (MOFA)[59] to identify factors that are correlated with CocUD consisting of features from both the ATAC and the RNA assay. We separated the ATAC assay by peak annotation to "distal" and "promoter" category as performed in Cell Ranger Arc to investigate the relationship between ATAC peaks in different genomic regions with RNA expression levels in the factor analysis. After the identification of highest variable features in the RNA ($N = 5000$), ATAC promoter ($N = 16,700$) and ATAC distal ($N = 42,651$) assays, a MOFA object based on 15 factors was trained in R using MOFA2 (v.1.8.0). In the resulting object, we tested for association of the resulting factors with CocUD status and performed MOFA-implemented downstream analyses such as gene set enrichment analysis (GSEA) and motif enrichment analyses in the CocUD-associated factors.

**Rank-Rank Hypergeometric Overlap.** We utilized the R package RRHO2[60] (v.1.074) to conduct rank-rank hypergeometric overlap (RRHO) analysis, aiming to assess convergent and divergent expression patterns at the transcriptome-wide level between male and female donors. RRHO scores were derived from comprehensive differential expression statistics from both male and female datasets. Differential expression statistics were calculated for comparisons with at least 40 cells per group. Subsequently, we evaluated overlapping signatures using the hypergeometric testing procedure implemented in RRHO2.

**GWAS enrichment analysis.** To evaluate the expression signatures of genes containing risk variants associated with substance use disorders including CocUD we performed a GWAS enrichment analysis using single-cell disease-relevance scoring (scDRS). Here, we aimed to identify cell types of the caudate nucleus that are enriched for the expression of risk genes and might therefore be especially susceptible to the effects of genetic risk variants. The following GWAS summary statistics for African American (AA) populations were included in the analysis: cocaine dependence $N = 3318$, Gelernter et al.[46], alcohol dependence[61] ($N = 6280$, Walters et al., 2018), cannabis use disorder[62] ($N = 9745$, Johnson et al., 2020), and opioid use disorder[63] ($N = 2555$, Polimanti et al., 2020). First, a gene-based analysis was performed for each summary statistics file using MAGMA[64] v.1.10. Next, scDRS was performed according with default settings as outlined in https://martinjzhang.github.io/scDRS/notebooks/

quickstart.html. First, a gene-$p$-value matrix was created including information on the strength of the association between the gene and the phenotype. The scdrs munge-gs function was applied to convert the GWAS-based gene statistics to scDRS format. Next, single-cell expression data was extracted from the Seurat object and converted to h5ad file-format. Finally, using the h5ad-file and the munged scDRS input file, disease-relevance scoring was performed using the compute-score function in scDRS. We used the scdrs perform-downstream function to evaluate group-level (i.e., cell cluster) associations with the inferred disease score.

## Validation

**3-crit model of cocaine use disorder and rodent tissue.** RT-qPCR was performed in tissue from the rat dorsal striatum, corresponding to the human caudate putamen, in male Sprague-Dawley rats from Charles River, Germany, that underwent the multisymptomatic 3-CRIT animal model of CocUD. Briefly, rats were implanted under isoflurane anesthesia with an intravenous catheter and allowed to recover for 7 days. Then, following extended intravenous self-administration (45 days), rats were tested on three measures modeled from criteria used to diagnose human use disorder in DSM-IV/V: the persistence of drug-seeking (responding when cocaine is signaled as unavailable), motivation for drug-seeking (progressive ratio), and drug taking despite adverse consequences (responding for cocaine in the presence of foot shock). The 3-CRIT model results in a distribution of animals ranging from those showing no addicted-like behavior (0-crit) to those showing addicted-like behavior on all three criteria (3-crit) (further described in Pohořalá, et al. [23]). For RT-qPCR in the current experiment, we analyzed tissue from six 0-crit and five 3-crit rats, as well as six control rats ($N_{total} = 17$) that were confined to the home cage for the entirety of experiment and thus had no exposure to cocaine or behavioral procedures. Two days following the completion of the 3-CRIT protocol, at 22 weeks of age, all rats were euthanized using isoflurane overdose followed by decapitation, at which point brains were extracted and flash-frozen in isopentane and stored at −80 °C until further processing. Experimental procedures were conducted according to the NIH ethical guidelines for the care and use of laboratory animals, in compliance with the German Animal Welfare Act, and approved by the local animal care committee (Regierungspräsidium Karlsruhe, Germany).

**Quantitative real-time PCR in the 3-crit model.** Total RNA was isolated from rat brain tissues using the RNeasy Mini Kit (Qiagen) and reverse transcribed into cDNA using the High-Capacity cDNA Reverse Transcription Kit (4368813; ThermoFisher). Quantitative real-time PCR (qPCR) was performed on the QuantStudio™ 7 Flex Real-Time PCR System (Applied Biosystems) with the following reaction setup: 10 µL TaqMan™ Gene Expression Master Mix (Applied Biosystems, catalog number: 4369016), 1 µL TaqMan™ Gene Expression Assay (Thermo Fisher Scientific), and 9 µL cDNA. Specific TaqMan™ Gene Expression Assays were used to detect the genes of interest: *PDE10A* (Rn00673152_m1), *KCNIP4* (Rn00710466_m1), *CACNA1C* (Rn00709287_m1), and *ZEB1* (Rn01538408_m1). The housekeeping gene *GUSB* (Rn00566655_m1) served as internal control. Gene expression analysis was performed in triplicates for the genes of interest and in duplicates for the internal control, with all samples analyzed together on the same plate per gene.

**Statistical analysis of 3-crit model qPCR data**
We generated the delta delta CT ($2^{-\Delta\Delta Ct}$) values by subtracting the Ct mean values of the triplicates of the genes of interest (*ZEB1, CACNA1C, PDE10A, KCNIP4*) from the Ct mean values of the duplicates of the internal control *GUSB*, then substracting the ΔCt from the average mean in the control group and potentiating this value with a base of 2. Group comparisons were performed between conditions using a Wilcoxon test.

**Reporting summary**

Further information on research design is available in the Nature Portfolio Reporting Summary linked to this article.

## Data availability

The datasets generated and analyzed during the current study are available in the European Genome Archive (EGA) under accession number EGAS50000000480.

## Code availability

The code used to analyze and visualize the dataset is deposited in the Github repository https://doi.org/10.5281/zenodo.14615181, https://github.com/lzillich/CN_multiome_cocaine.

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

## Acknowledgements

We are grateful for the invaluable donations and participation from families, as well as for the generous collaboration of the medical examiners at the Harris County Institute of Forensic Sciences. We also thank all colleagues who have contributed to the UTHealth Brain Collection and whose work continues to grow and broaden the collection. Postmortem tissue was sampled based on written informed consent from each donor's next of kin. The study was conducted in line with the guidelines of the Declaration of Helsinki and was approved by the Ethics Committee II of the Medical Faculty Mannheim under the register number 2021-681 and by the University of Texas Health Science Center at Houston (HSC-MS-15-0247). Financial support for this work was provided by the Deutsche Forschungsgemeinschaft (DFG, German Research Foundation)—Project-ID 402170461 (B10) – TRR 265[65] to S.H.W., A.M.M.O., and R.E.B., and Project-ID 433081111 to P.K. The study was funded by the German Federal Ministry of Education and Research (BMBF) through the e:Med consortium "A systems-medicine approach towards distinct and shared resilience and pathological mechanisms of substance use disorders - SysMedSUDs" (01ZX01909 to R.S., P.K., M.R., A.C.H., S.H.W.). FS is supported by a Families for Borderline Personality Disorder Research (Beth and Rob Elliott) 2023 NARSAD Young Investigator Grant (# 31537). R01DA044859 and the John S. Dunn Foundation to C.W.B.. A.M.M.O is supported by a Heisenberg grant from the DFG (OL 437/3). We thank Joel Gelernter and his group for sharing the summary statistics of the cocaine dependence GWAS[46]. Figure 1A (bottom right) and 4D (upper) were created with BioRender.com. P.K., L.Z., A.A., A.J., and J.L. acknowledge the generous support of the Hector Stiftung II. For the publication fee we acknowledge financial support by Heidelberg University.

## Author contributions

L.Z., S.H.W., P.K., and R.S. conceptualized the project. L.Z. managed the project. L.Z. and E.Z. performed single-nuclei data analysis for the project. S.H.W., M.R., A.C.H., R.S., A.M.M.O., and R.E.B. obtained funding for the project. T.D.M., L.S., and C.W.-B. collected human postmortem brain tissue and performed psychological autopsies, and L.S. dissected the tissue for the project. A.A. and A.J. conducted the single nuclei extraction and single nuclei library preparations. V.P. and R.E.B. conducted all 3-CRIT experiments and dissected the tissue. H.B., A.M.M.O., and V.P. performed qPCR experiments and data analysis. L.Z. wrote the original manuscript with figures. L.Z., E.Z., A.A., V.P., L.S., H.B., A.J., J.F., F.S., D.A., M.P.V., S.M., A.C.H., T.D.M., M.R., J.L., R.S., A.M.M.O., C.W.-B., R.E.B., P.K., and S.H.W. participated in reviewing and editing the manuscript for publication.

## Funding

## Competing interests

The authors declare no competing interests.

## Additional information

**Supplementary information** The online version contains
supplementary material available at

Lea Zillich.

**Peer review information** *Nature Communications* thanks Bo Zhang and
the other, anonymous, reviewer(s) for their contribution to the peer
review of this work. A peer review file is available.

