## [Transparent Peer Review file · Nature Communications]

Cell type-specific Multi-Omics Analysis of Cocaine Use Disorder in the Human Caudate Nucleus

Corresponding Author: Dr Lea Zillich

Version 0:

Reviewer comments:

Reviewer #1

(Remarks to the Author)

The authors present a single-nuclei multiome analysis of postmortem caudate from African American individuals who had cocaine use disorder. The dorsal striatum makes sense as the focus given its role in addictive behavior. Cocaine is a relevant substance, especially in the African American community. Single cell multiome data from brain tissue is quite rare and doubly so from human postmortem brain. Overall there is a lot to like about the basis of this study and the manuscript is also well written with a few exceptions noted below. There are several major issues however with the single nuclei data and analysis.

Although the sample size is acceptable for a study of human postmortem brain, the number of cells per sample is fairly low. The relatively small cell counts makes it difficult to effectively analyze less common types of cells. Sample size and cell count are also an issue for any sex based analyses since only two women with cocaine use disorder are included and they represent a tiny fraction of total cells.

The authors used a Wilcoxon rank sum test for differential expression analysis and that test is not conservative enough for analyzing single cell expression data because it does not account for differences between samples. The test is therefore prone to false positives. This issue is made worse due to the presence of two notable outlier samples that contribute half of the cells in the data set and almost 80% of the cocaine use disorder cells. Any findings in the Wilcoxon rank sum test are likely to be driven almost entirely by these two samples and the generalizability of those findings to CUD in general is unclear.

The processing of the data does not include any adjustments for ambient RNA or systematic doublet removal with packages like DoubletFinder or scDbfFinder. This further increases the possibility of false positives from the Wilcoxon test. In total these problems with the data processing and analysis make it difficult to have much confidence in the differential expression results and all of the additional analyses that are downstream of them.

A number of more minor issues are also present in the manuscript. It is not clear why the authors chose to analyze the delta Ct from the qPCR instead of the more standard $2^{-\Delta\Delta Ct}$ method.

The results mention the strongest deregulation of expression occurring in astrocytes and medium spiny neurons but the numbers of differentially expressed genes noted there do not match the DE genes portion of Figure 2B.

KCNIP4 is included in the qPCR of rat caudate due to the importance in GO terms but that importance is never explained.

GABAergic CALB2 cells are incorrectly referred to as MSNs in the results.

The significant results in 0-crit but not 3-crit animals are interesting and they should be discussed in more detail in the discussion. The authors correctly note that exposure and substance use disorder cannot truly be parsed in humans with cocaine use disorder but the rat results certainly imply where these genes might be relevant.

More context would be helpful for the distinct D1 MSN ADARB2 population. Some attempt should be made to harmonize that population with the human dorsal striatum clustering in Phan et al (Nature Communications, 2023).

The upset plot in Figure 2B appears to be slightly broken. At least on the available reviewer PDF the axes and labels are not in the correct place.

Figure 2C has a set of overlapping numbers in the bottom left that are unreadable.

(Remarks on code availability)

The R and R Markdown files in the Github are sufficient for the community to vet the analyses in the manuscript and use as a resource for future work.

Reviewer #2

(Remarks to the Author)

This study performed a multi-omic study of cocaine use disorder (CocUD) in human postmortem tissue from the caudate nucleus. As stated by the authors, this is the first human study of CocUD at single-nuclei resolution using a multi-omic approach (ATAC-seq and RNA-seq). The manuscript is clearly organized and well-written, and the findings are of importance to the field – though the importance of these findings is not always made clearly apparent. My one major concern about the manuscript in its current form is that the discussion reads as a summary of the many analyses that were conducted. While some of this summary is needed, it was hard to know which findings were important and why they were important and how to interpret them within the context of what we already know about CocUD and SUD more broadly. As one example, in the summary paragraph of the discussion the authors state that they have novel findings for CocUD, however, based on the current discussion the only one that is highlighted is ZEB1.

Minor:

- Cocaine use Disorder is sometimes abbreviated as CocUD and sometimes as CUD. The CocUD abbreviation is preferred so as not to be confused with cannabis use disorder. Either way, consistent use of abbreviation is needed.
- There are many acronyms used throughout the manuscript and, in some spots, the overuse of these acronyms makes reading the manuscript difficult. Is it possible to either reduce the number of acronyms used or provide cheat sheet for all/most the acronyms?

(Remarks on code availability)

Reviewer #3

(Remarks to the Author)

Zillich and colleagues investigated the changes of molecular signatures underlying cocaine use disorder by performing single-nuclei multiome profiling on postmortem caudate nucleus tissue from individuals with CocUD and controls. Their analysis identified 13 cell types, revealing 1,383 differentially regulated genes and significant alterations in medium spiny neurons (MSNs) and astrocytes related to neurotransmitter activity and synapse organization. Key findings included the transcription factor ZEB1's role in CocUD-specific subclusters and the identification of PDE10A as a potential drug target, highlighting cell type-specific molecular changes and the value of multi-omics approaches in addiction research. The results clearly indicated transcriptomic and epigenetic changes under cocaine use in the human brain and provided unique data resources and important findings to better understand the cocaine use disorder at the single cell level. However, there are certain concerns that should be addressed for a better and clearer understanding of the presented work to readers.

1. Overall, the authors need to better describe the results presented in figures to help readers understand this work, which currently seems without a very rough, careful checking and polishing. The results sections are simple, and the figure legends also very briefly describe the figures. Figure 2C/D, what does the area of each section mean? The edge? No color bar in Figure 2G. Does color mean p-value in Figure 3B? Figure 3C, the meaning of 2 colors of gene name? Figure 3D, meaning of color? Figure 3H, TPM for expression? Figure 4A, meaning of color and size of nodes?
2. An upset plot is needed to indicate the overlaps among DARs of each cell type.
3. TF motifs are highly similar within the family. It is not clear how the author checked the redundancy of motif prediction. Thus, the predicted activation and repression function of ZEB1 described in results need more evidence and analysis.
4. The linked peaks-gene pairs showed a relatively unmatched pattern. In D1 MSN ADARB2, the gene expression did not correlate to ATAC signal changes. In D1 MSN, D2 MSN, and OPC, the expression changes seem reversed to ATAC changes. The author should explain the correlation for these defined pairs. Moreover, it is not clear how authors defined common and specific, and what's the relationship among these parties to DEG and DARs? How such specific information was used in the downstream analysis?

5. It is not clear the relationship between Figure 3A and Figure 3E?
6. In Fig4D, why did the author not use the double delta CT method?
7. the description of MOFA is not clear. How do the factors identified in this analysis connect to CoCUD besides some similar enriched GO terms? How do we explain the negative enrichment ZEB1 in factor 8 gene set?
8. The descriptions of sex-DE analysis and GWAS enrichment are too simple to understand. Especially for negative results of GWAS enrichment at the cell type level.

(Remarks on code availability)

Version 1:

Reviewer comments:

Reviewer #2

(Remarks to the Author)

Previous concerns were adequately addressed. I have no further comments at this time.

(Remarks on code availability)

N/A

Reviewer #3

(Remarks to the Author)

I appreciate authors addressed my comments, I don't have more concerns.

(Remarks on code availability)

A detailed readme on GitHub should be available to the deposited files.

Rebuttal Letter - Cell type-specific Multi-Omics Analysis of Cocaine Use Disorder in the Human Caudate Nucleus

We thank the reviewers for their constructive comments and suggestions. We are convinced that the changes based on the comments and suggestions have strengthened our manuscript substantially and hope that our work will now be considered suitable for publication in Nature Communications. Our point-by-point responses to the reviewer comments are provided below, with the following formatting:

Original reviewer comments are in **bold**.

Our responses are in regular text.

Quotations from the manuscripts are in *italics*.

Changes to existing manuscript text are indicated in blue (both in this files and the manuscript file).

Reviewer #1 (Remarks to the Author):

The authors present a single-nuclei multiome analysis of postmortem caudate from African American individuals who had cocaine use disorder. The dorsal striatum makes sense as the focus given its role in addictive behavior. Cocaine is a relevant substance, especially in the African American community. Single cell multiome data from brain tissue is quite rare and doubly so from human postmortem brain. Overall there is a lot to like about the basis of this study and the manuscript is also well written with a few exceptions noted below. There are several major issues however with the single nuclei data and analysis.

Comment 1.1: Although the sample size is acceptable for a study of human postmortem brain, the number of cells per sample is fairly low. The relatively small cell counts makes it difficult to effectively analyze less common types of cells. Sample size and cell count are also an issue for any sex based analyses since only two women with cocaine use disorder are included and they represent a tiny fraction of total cells.

Response 1.1: We agree with the reviewer that a larger sample size would be ideal. The number of retained cells was indeed smaller than we expected based on our initial tests. We have since revised our nuclei extraction protocol to achieve a higher yield. We now explicitly mention the number of cells per sample in the limitation section.

Regarding the sex-based analyses, we faced the issue of whether not to perform an analysis and assume that there are no differences between the sexes or perform one with limited interpretability, because of the small sample size. In the end, we decided to perform the analysis, mainly because of the underrepresentation of females in the substance use disorder literature and to stimulate potential future analyses with larger sample sizes. At the same time, we decided to present this analysis as an exploratory posthoc analysis and now only include results for the six largest cell types, with at least 40 cells per group (e.g. N= 44 for Microglia from female donors with cocaine use disorder). Similar to the cell counts per sample, we included the limited interpretability of the sex-based analysis in the discussion section.

We have modified the methods section (lines: 577-584):

Rank-Rank Hypergeometric Overlap

We utilized the R package RRHO2⁵⁹ (v.1.074) to conduct rank-rank hypergeometric overlap (RRHO) analysis, aiming to assess convergent and divergent expression patterns at the transcriptome-wide level between male and female donors. RRHO scores were derived from comprehensive differential expression statistics from both male and female datasets. Differential expression statistics were calculated for comparisons with at least 40 cells per group. Subsequently, we evaluated overlapping signatures using the hypergeometric testing procedure implemented in RRHO2.

And discuss the results in more detail (line: 392-397):

At the same time, the sex-specific effects need to be interpreted with caution. While the single-nuclei framework allowed for a high statistical power, because each cell was treated as one observation, the overall sample size was still small and results could lack generalizability. Larger studies are needed to conclusively investigate sex-specific effects in CocUD. Similarly, the overall sample size in the current study and the number of nuclei per cell type remain a limitation that requires more data generation and integration.

Comment 1.2: The authors used a Wilcoxon rank sum test for differential expression analysis and that test is not conservative enough for analyzing single cell expression data because it does not account for differences between samples. The test is therefore prone to false positives. This issue is made worse due to the presence of two notable outlier samples that contribute half of the cells in the data set and almost 80% of the cocaine use disorder cells. Any findings in the Wilcoxon rank sum test are likely to be driven almost entirely by these two samples and the generalizability of those findings to CUD in general is unclear.

Response 1.2: We agree with the reviewer that using the Wilcoxon rank sum test is not ideal and can lead to false positives. At the same time, the strength of single-nuclei sequencing is to observe the transcriptome of each individual cell and maintain this variance. We considered pseudobulking, but agreed that this limits the strength of the data immensely and would lead to an experiment that could have also been performed as a bulk RNA-seq study with FACS-sorted nuclei. In our opinion, an ideal statistical framework would be a hierarchical regression, which our sample size unfortunately does not allow for. We are aware of this limitation. Our main arguments for calculating the Wilcoxon rank sum tests despite the limitations are that we observe very similar cell type distributions across samples and homogeneous cell clusters (Figure 1E, Sup. Figure S1), we also only consider DE genes with a log₂FC of at least 0.25 and if there were expressed in at least 25% of cells. While there is still a possibility of false positives, we are confident in our results, because we observe convergent results across multiple different methods, such as MOFA. Also, eliminating about 1,000 cells classified as doublets still produced the same results (see also response 1.3). We hope the reviewer agrees with this approach. We also included it in the discussion as a potential limitation (line 397-400):

Larger sample sizes would also allow for using state-of-the-art statistical methods, such as hierarchical modeling, less susceptible to false positives than the Wilcoxon rank sum test, while maintaining the single-cell resolution.

Comment 1.3: The processing of the data does not include any adjustments for ambient RNA or systematic doublet removal with packages like DoubletFinder or scDbfFinder. This further

increases the possibility of false positives from the Wilcoxon test. In total these problems with the data processing and analysis make it difficult to have much confidence in the differential expression results and all of the additional analyses that are downstream of them.

Response 1.3: We thank the reviewer for this comment. We agree that it is important to adjust the data for doublets and, if necessary, for ambient RNA. Our initial analysis indicated a small number of doublets per sample. For most samples, these were evenly distributed across feature counts, although one would expect that doublets contain more features than singlets on average. Therefore, we did not exclude doublets but cells with high feature counts. However, we agree with the reviewer that we should mitigate the risk of an increasing number of false positives in the Wilcoxon test and now remove doublets for each sample. Therefore, we repeated the entire workflow and the differing number of cells does lead to slight changes in the results throughout the manuscripts, which is why we updated all tables and figures. In some cell types the order of DE genes has shifted, e.g. MBP is now the third most significant upregulated gene not the first. However, the results and their interpretation has not changed. The most noticeable difference is that the correlation between microglia-MOFA factor 5 and cocaine use disorder is significantly lower in this version, which is why we do not include factor 5 in the results anymore.

Regarding the ambient RNA, we saw no evidence during the quality control that indicated contamination with ambient RNA. For example, in other experiments, where we used SoupX to adjust for ambient RNA, we observed many cells with low gene counts, marker profiles that were difficult to interpret, and what 10X refers to as “a missing knee” in the cellranger web summary. Fortunately, this was not the case in the present study. In addition, algorithms like SoupX directly adjust the count matrix, which can potentially introduce bias as well. We therefore decided to not adjust for ambient RNA in this study and hope the reviewer agrees with our reasoning.

We have now included the DoubletFinder analysis and report this in the methods as follows (line 445-451):

In Seurat, quality control was performed and nuclei were excluded if they did not meet the following parameters: number of RNA features between 200 and 6,500, number of ATAC features between 1,000 and 25,000, percentage of mitochondrial genes < 10%, TSS enrichment score between 2 and 8, and nucleosome signal between 0.5 and 2. We used DoubletFinder⁴⁶ to identify doublets, two cells that were captured in the same GEM, and removed the doublets for each sample (Supplementary Table S1).

Comment 1.4: A number of more minor issues are also present in the manuscript. It is not clear why the authors chose to analyze the delta Ct from the qPCR instead of the more standard $2^{-\Delta\Delta Ct}$ method.

Response 1.4: We thank the reviewer for this suggestion and now use the $2^{-\Delta\Delta Ct}$ method. During this revision, we noticed a coding error that led to a wrong direction of effect in CACNA1C in the original version, which we have now corrected. All other qPCR results did not change. The methods and results sections now read as follows:

Line 641-646:

Statistical Analysis of 3-crit model qPCR data

We generated the delta delta CT ($2^{-\Delta\Delta Ct}$) values by subtracting the Ct mean values of the triplicates of the genes of interest (ZEB1, CACNA1C, PDE10A, KCNIP4) from the Ct mean values of the duplicates of the internal control GUSB, then subtracting the ΔCt from the average mean in the control group and potentiating this value with a base of 2. Group comparisons were performed between conditions using a Wilcoxon test.

Line 255-261:

We observed differential expression in PDE10A ($p=0.015$), consistent with an upregulation in the non-neuronal cell types (Supplementary Table 2), and in CACNA1C ($p=0.026$), in line with an upregulation in microglia and oligodendrocytes, while a downregulation was observed in MSNs in the DE analysis (Supplementary Table S2). These effects were significant for 0-crit, but not 3-crit animals, compared to controls, indicating either a cocaine-consumption effect or a habituation in the 3-crit animals. No effect was observed for ZEB1 ($p=0.82$) and KCNIP4 ($p=0.82$).

Comment 1.5: The results mention the strongest deregulation of expression occurring in astrocytes and medium spiny neurons but the numbers of differentially expressed genes noted there do not match the DE genes portion of Figure 2B.

Response 1.5: We believe that this misunderstanding stems from us listing the number of unique and shared differentially expressed genes and not the total number in the text. We have now corrected this as follows:

Line 133-136:

Here, we identified 1,485 differentially expressed (DE) genes (Sup. Table S2), clustering distinctly according to glial or neuronal cell identity (Fig. 2A). Strongest deregulation of gene expression levels was observed in astrocytes (368 unique DE genes) and medium spiny neurons (D1: 20, D2: 80, D1/D2: 201 unique DE genes).

Comment 1.6: KCNIP4 is included in the qPCR of rat caudate due to the importance in GO terms but that importance is never explained.

Response 1.6: We thank the reviewer for mentioning this. KCNIP4 is involved in many of the synaptic signaling-related GO terms implicated in the differential expression analyses, MOFA, and Pando. We have rephrased our argument for choosing the genes for validation and also include an additional table in the Supplementary Information listing the GO terms and the genes involved.

Line 246-248:

We prioritized the DE genes from the CocUD DE analyses based on their relationship with the DA peaks (CACNA1C), their involvement in the synaptic signaling GO terms (KCNIP4; Sup. Table S8), their role in the GRN (ZEB1) and the druggability analysis (PDE10A).

Comment 1.7: CALB2 cells are incorrectly referred to as MSNs in the results.

Response 1.7: We thank the reviewer for pointing this typo out. We now present only sex-specific results for larger cell types and therefore deleted this line altogether (see also response 1.1).

Comment 1.8: The significant results in 0-crit but not 3-crit animals are interesting and they should be discussed in more detail in the discussion. The authors correctly note that exposure and substance use disorder cannot truly be parsed in humans with cocaine use disorder but the rat results certainly imply where these genes might be relevant.

Response 1.8: We agree with the reviewer that this is an important point that warrants further discussion. We now included the following paragraph in the discussion:

Line 363-378:

We explored whether broad gene expression changes of GRN-identified targets were observed in the 3-crit model of CocUD. Here, we validated the effects for PDE10A and CACNA1C in 0-crit, but not 3-crit animals relative to naïve controls. This finding could suggest that alterations in PDE10A and CACNA1C may be due to long-term cocaine exposure rather than CocUD, per se, and highlights an important challenge in addiction research: based on postmortem human brain tissue alone, it is difficult to conclude whether the observed effects result from substance intake or reflect neuroadaptations resulting from substance use disorder. However, because 0- and 3-crit animals don't differ in cocaine exposure, it is likely that differences between these two groups in PDE10A and CACNA1C relative to controls reflect neuroplastic/neuroadaptive changes in response to an addiction-like phenotype not seen in the human samples. Furthermore, the 3-CRIT model assesses three specific criteria of CocUD in humans that are modelled for rats²², such that this model does not perfectly represent the diverse symptomology of CocUD in humans. Nonetheless, our study demonstrates some convergence among CocUD in humans and a rat model, and represents a unique strategy for potential validation of targets identified by multi-omics from human tissue.

Comment 1.9: More context would be helpful for the distinct D1 MSN ADARB2 population. Some attempt should be made to harmonize that population with the human dorsal striatum clustering in Phan et al (Nature Communications, 2023).

Response 1.9: While our D1 ADARB2 cluster showed similar marker expressions as the hybrid D1/D2 MSN cluster in Phan et al. (2023), our cluster shows little to no DRD2 expression and we therefore did not consider them hybrid MSNs. However, we now include a statement about the expression profile compared to the hybrid MSN cluster from Phan et al. (2023) in the results and methods sections:

Results (line 125-128):

Annotation of clusters was supported by the expression of known marker genes (Fig. 1F), as well as their promoter accessibility (Fig. 1G) and cross-referenced with a recent publication profiling the dorsal striatum in opioid use disorder²⁶.

Methods (line 468-481):

The expression of known marker genes for cell types in the human striatum⁴⁹ and different types of expected interneurons was evaluated in the RNA assay. In addition, we utilized the FindAllMarkers function to generate differential expression profiles distinguishing expression in one cluster from all other clusters (Supplementary Fig. 5). We cross-checked highly expressed genes in clusters with the Transcriptomics Explorer of the Allen Human Brain Atlas⁵⁰ and the Human Protein Atlas⁵¹ leading to the identification of 14 different cell type clusters. One minor cluster was characterized by strong expression of mitochondrial genes as evaluated by the FindAllMarkers function in R. We excluded this cluster expressing oligodendrocyte markers from the object and performed further analysis in the 13-cluster object. In addition, we compared our marker expression profile to the human dorsal striatum dataset described in Phan et al. (2023)²⁶ and observed similarities between our D1-MSN-ADARB2 cluster and the D1/D2-MSN hybrid cluster described in Phan, et al. ²⁶. At the same time, we only observed spurious DRD2 expression in our D1-MSN-ADARB2 cluster suggesting them as a D1-MSN subpopulation.

Comment 1.10: The upset plot in Figure 2B appears to be slightly broken. At least on the available reviewer PDF the axes and labels are not in the correct place.

Response 1.10: We thank the reviewer for pointing this out. We identified the source of the glitch and updated Figure 2B accordingly.

Comment 1.11: Figure 2C has a set of overlapping numbers in the bottom left that are unreadable.

Response 1.11: We thank the reviewer for noticing and fixed Figure 2C (now 2D).

Reviewer #2 (Remarks to the Author):

Comment 2.1: This study performed a multi-omic study of cocaine use disorder (CocUD) in human postmortem tissue from the caudate nucleus. As stated by the authors, this is the first human study of CocUD at single-nuclei resolution using a multi-omic approach (ATAC-seq and RNA-seq). The manuscript is clearly organized and well-written, and the findings are of importance to the field – though the importance of these findings is not always made clearly apparent. My one major concern about the manuscript in its current form is that the discussion reads as a summary of the many analyses that were conducted. While some of this summary is needed, it was hard to know which findings were important and why they were important and how to interpret them within the context of what we already know about CocUD and SUD more broadly. As one example, in the summary paragraph of the discussion the authors state that they have novel findings for CocUD, however, based on the current discussion the only one that is highlighted is ZEB1.

Response 2.1: We thank the reviewer for this important comment. In the original version, we tried to focus the discussion on the most exciting results, from our perspectives. We agree that other results were not discussed with adequate depth. Next to the most influential transcription factor, *ZEB1*, we now discuss another one of the highly connected and validated downstream targets, *CACNA1C*. We also provide a more in-depth discussion of the validation experiments and discuss additional limitations as raised by reviewer 1.

Discussion, line 353-378:

A highly connected downstream target in the GRN with large effects in the DE analysis was CACNA1C, calcium voltage-gated channel subunit alpha1 C, which has been broadly implicated in psychiatric disorders³⁹. Further, there is evidence for an important role of CACNA1C in cocaine addiction. In a rodent model of CocUD it has been shown that CACNA1C heterozygous mice demonstrate an increased reward response to cocaine⁴⁰. In another study, L-type calcium channel blockage by nifedipine was associated with less neurotoxicity in the prefrontal cortex of rats exposed to acute cocaine⁴¹. At the same time, we observed a downregulation of CACNA1C in the DE analysis in MSNs and an upregulation in cocaine-exposed animals in the 3-crit model, emphasizing the importance of distinguishing acute and long-term exposure and cell type-specific effects.

We explored whether broad gene expression changes of GRN-identified targets were observed in the 3-crit model of CocUD. Here, we validated the effects for PDE10A and CACNA1C in 0-crit, but not 3-crit animals relative to naïve controls. This finding could suggest that alterations in PDE10A and CACNA1C may be due to long-term cocaine exposure rather than CocUD, per se, and highlights an important challenge in addiction research: based on postmortem human brain tissue alone, it is difficult to conclude whether the observed effects result from substance intake or reflect neuroadaptations resulting from substance use disorder. However, because 0- and 3-crit animals don't differ in cocaine exposure, it is likely that differences between these two groups in PDE10A and CACNA1C relative to controls reflect neuroplastic/neuroadaptive changes in response to an addiction-like phenotype not seen in the human samples. Furthermore, the 3-CRIT model assesses three specific criteria of CocUD in humans that are modelled for rats²², such that this model does not perfectly represent the diverse symptomology of CocUD in humans. Nonetheless, our study demonstrates some convergence among CocUD in humans and a rat model, and represents a unique strategy for potential validation of targets identified by multi-omics from human tissue.

Minor:

Comment 2.2: Cocaine use Disorder is sometimes abbreviated as CocUD and sometimes as CUD. The CocUD abbreviation is preferred so as not to be confused with cannabis use disorder. Either way, consistent use of abbreviation is needed.

Response 2.2: We thank the reviewer for pointing this out. We corrected the abbreviations.

Comment 2.3: There are many acronyms used throughout the manuscript and, in some spots, the overuse of these acronyms makes reading the manuscript difficult. Is it possible to either reduce the number of acronyms used or provide cheat sheet for all/most the acronyms?

Response 2.3: Thank you for this important comment. We reduced two acronyms, immediate early genes, and overrepresentation analysis. More importantly, we now provide a full table of all acronyms (line 47-48).

Acronyms

CocUD	Cocaine use disorder
Ctrl	Control
DA	Differentially accessible
DE	Differentially expressed
GO	Gene ontology
GRN	Gene regulatory network
KEGG	Kyoto Encyclopedia of Genes and Genomes
MOFA	Multi Omics Factor Analysis
MSN	Medium spiny neurons
OPC	Oligodendrocyte precursor cell
scDRS	Single-cell disease relevance score
TF	Transcription factor

Reviewer #3 (Remarks to the Author):

Zillich and colleagues investigated the changes of molecular signatures underlying cocaine use disorder by performing single-nuclei multiome profiling on postmortem caudate nucleus tissue from individuals with CocUD and controls. Their analysis identified 13 cell types, revealing 1,383 differentially regulated genes and significant alterations in medium spiny neurons (MSNs) and astrocytes related to neurotransmitter activity and synapse organization. Key findings included the transcription factor ZEB1's role in CocUD-specific subclusters and the identification of PDE10A as a potential drug target, highlighting cell type-specific molecular changes and the value of multi-omics approaches in addiction research. The results clearly indicated transcriptomic and epigenetic changes under cocaine use in the human brain and provided unique data resources and important findings to better understand the cocaine use disorder at the single cell level. However, there are certain concerns that should be addressed for a better and clearer understanding of the presented work to readers.

Comment 3.1. Overall, the authors need to better describe the results presented in figures to help readers understand this work, which currently seems without a very rough, careful checking and polishing. The results sections are simple, and the figure legends also very briefly describe the figures. Figure 2C/D, what does the area of each section mean? The edge? No color bar in Figure 2G. Does color mean p-value in Figure 3B? Figure 3C, the meaning of 2 colors of gene name? Figure 3D, meaning of color? Figure 3H, TPM for expression?

Figure 4A, meaning of color and size of nodes?

Response 3.1: We thank the reviewer for raising this issue and now provide more detailed figure legends (line 647-708).

Figure 1. A) sample workflow for $N=14$ individuals and bioinformatics workflow relating each analysis to the corresponding figure. B) integrated UMAP for 30,030 nuclei, depicting 13 clearly delineated cell types. C) UMAP of the RNA assay. D) UMAP of the ATAC assay. E) cell type proportions per sample. F) Dot plot depicting marker gene expression per cell type. G) coverage plot depicting chromatin peaks in promoter regions of cell type marker genes. CN = caudate nucleus, CocUD = cocaine use disorder, UMAP = uniform manifold approximation and projection.

Figure 2. A) Heatmap depicting differentially expressed genes (adjusted p -value <0.05 , $\log_2FC > 0.5$, expressed in at least 25% of cells), red = upregulated in CocUD, blue = downregulated in CocUD, top ten DE genes per cell type are labeled. Upset plot depicting the number of shared and distinct differentially expressed features per cell type, for B) differential expression and C) differential accessibility. D) emapplot depicting the top 14 category results of the Gene Ontology overrepresentation analysis for DE genes in a cell type-specific manner. Each circle represents one GO term as a pie chart, indicating the strength of enrichment in the different cell types, represented by color. Edges indicate semantically similar GO terms, with overlapping genes. Circle size indicates the number of genes in the pathway. E) emapplot depicting the top 14 category results of the Kyoto Encyclopedia of Genes and Genomes (KEGG) overrepresentation analysis for DE genes in a cell type-specific manner. Each circle represents one GO term as a pie chart, indicating the strength of enrichment in the different cell types, represented by color. Edges indicate semantically similar GO terms, with overlapping genes.

Circle size indicates the number of genes in the pathway. F) heatmap showing differentially accessible (DA) peaks (adjusted p -value <0.05 , $\log_2FC > 0.25$, expressed in at least 5% of cells), red = upregulated in CocUD, blue = downregulated in CocUD. G) Annotation of DA peaks to their genomic location. H) comparison between genomic location of DA peaks to the genomic background, *** = $p < 0.001$, ** = $p < 0.01$, * = $p < 0.05$.

Figure 3. A) Heatmap depicting shared (brown), CocUD-specific (red), and control-specific (blue) links between chromatin accessibility and RNA expression by cell type. B) emapplot showing overrepresented gene ontology biological processes for genes characterized by a CocUD-specific linked peak in A), color represents statistical significance and circle size the number of genes in the GO term. C) heatmap depicting cell type-specific motif enrichment, yellow motif names indicate increased activity while motif names in blue represent reduced activity of the motif in cell types based on chromVAR activity scoring. D) Feature plots confirming motif expression for MA1114.1/PBX3 in D1- and D2-MSNs, MA1577.1/TLX2 in astrocytes and MA0080.5/SPI1 in Microglia. Enrichment scores are color-coded from blue (no enrichment) to yellow (strong enrichment). E) Heatmap depicting differential transcription factor motif activity between CocUD and controls, red = increased activity in CocUD, blue = decreased activity in CocUD. F) Heatmap depicting corresponding transcription factor differential transcript expression in CocUD, red = upregulated in CocUD, blue = downregulated in CocUD. G) transcription factor motifs for ZEB1 and FOXP2, size of sequence represents probability. H) feature plots depicting ZEB1 and FOXP2 expression in cell types of the caudate nucleus (log-normalized counts).

Figure 4. A) Gene Regulatory Network (GRN) from the D1- and D2-MSNs, color and size of nodes relate to centrality in the network, larger circles represent increased centrality, orange edges represent activation, and grey edges inhibition. B) Feature plots depicting ZEB1 RNA expression (upper), module scores for ZEB1 suppressed downstream targets (middle), and module scores for ZEB1 activating downstream targets (lower), separately for CocUD (left) and controls (right). C) emapplot summarizing GO overrepresentation results from genes activated (red) and suppressed (turquoise) by ZEB1, PBX3, and ZNF148, the top TFs from regulon scoring. Each circle represents one GO term as a pie chart, indicating the strength of enrichment in the different regulons, represented by color. Edges indicate semantically similar GO terms, with overlapping genes. Circle size indicates the number of genes in the pathway. D) summary of validation analysis in 3-crit model of cocaine use disorder in rats, light grey = controls (N=6), grey = 0crit (N=6), black = 3crit (N=5), * = $p < 0.05$. E) RRHO plots showing the overlap between DE results in males and females, upper left = upregulated in females and downregulated in males, upper right = upregulated in males and females, lower left = downregulated in males and females, lower right = upregulated in males and downregulated in females. MSNs=medium spiny neurons, GRN= gene regulatory network, CocUD=cocaine use disorder, ctrl = control, (+) = activating, (-) = suppressing.

Comment 3.2: An upset plot is needed to indicate the overlaps among DARs of each cell type.

Response 3.2: We agree with the reviewer that an upset plot depicts a concise visualization to investigate DAR overlap across cell types. We now show this upset plot in Figure 2C and report on the results in the section “Differential Expression & Accessibility Analysis” (line: 131-158).

Differential accessibility analysis revealed 10,342 differentially accessible (DA) peaks (Sup. Table S3), again with a clear distinction between glial and neuronal cell types (Fig. 2F). The highest number of DA peaks was observed in astrocytes and oligodendrocytes (Figure 2C).

Comment 3.3: TF motifs are highly similar within the family. It is not clear how the author checked the redundancy of motif prediction. Thus, the predicted activation and repression function of ZEB1 described in results need more evidence and analysis.

Response 3.3: We thank the reviewer for this comment and agree that redundancy depicts a known problem in motif prediction. To address this issue, when running chromVAR and Pando, we used motif information from the JASPAR2020 position frequency matrix (PFM). In comparison to other motif databases such as TRANSFAC, JASPAR2020 provides non-redundant motif information based on experimentally determined DNA binding sequences (<https://jaspar2020.genereg.net/docs/>). Thus, we aimed to reduce redundancy by using an input dataset that provides non-redundant motif information. We have added this information to the methods section

Motif enrichment analysis (line 528-537)

To assess the enrichment of transcriptional motifs within DA peaks associated with CocUD status, we performed motif enrichment analysis (MEA) using chromVAR51 (v.1.20.2). The JASPAR2020 (<http://jaspar.genereg.net/>) position frequency matrix was used to obtain a comprehensive and experimentally validated set of non-redundant human transcription factor motifs.

Identification of gene regulatory networks (line 540-548)

After GRN initiation and motif scanning based on non-redundant human JASPAR2020 TF binding motifs, transcription factor modules were identified using the following parameters:

p-value threshold = 0.05, number of variables = 5, minimum number of genes per module = 5, and R2=0.1.

Comment 3.4: The linked peaks-gene pairs showed a relatively unmatched pattern. In D1 MSN ADARB2, the gene expression did not correlate to ATAC signal changes. In D1 MSN, D2 MSN, and OPC, the expression changes seem reversed to ATAC changes. The author should explain the correlation for these defined pairs. Moreover, it is not clear how authors defined common and specific, and what's the relationship among these parties to DEG and DARs? How such specific information was used in the downstream analysis?

Response 3.4: We agree with the reviewer that the peak-gene linkage pattern appears to be unmatched for some cell types. We hypothesize several reasons for this observation. The peaks displayed in the heatmap are peaks with differential accessibility in CocUD ($\text{abs}(\log_2\text{FC}) > 0$, $p.\text{adjust} < 0.05$) that displayed at least one significant peak-gene association in the links analysis. For all genes, we selected for the strongest absolute correlation coefficient of a peak-gene association pair. Since we selected for absolute values of correlation, reverse expression changes might be observable in the heatmap. Further, as only the largest correlation coefficient is shown for each gene, there might be further peaks associated with a gene with inverse correlation patterns that might then in sum contribute to the overall effect on expression. Even a true inverse correlation pattern might make sense at the biological level in case the region with stronger accessibility contains a motif for a TF with repressive activity. Thus, with Figure 3A we aimed to provide a descriptive overview on peak-gene relationships in our dataset that we then investigated in a targeted approach in follow-up analyses such as Pando where we also included DE gene and motif information.

We further agree that the definition of common and specific links should be further elaborated. We have added an additional methods section on “Linking peaks to genes” where we describe the definition and calculation of peak-gene correlations (line 517-525).

Linking peaks to genes

Peak-gene associations were identified using the LinkPeaks functionality of Signac. Here, the correlation between accessibility of peaks within a threshold window of max 500kb distance from transcription start site and the expression level of respective genes is determined. To identify peak-gene correlations specific to either CoCUD or Ctrl, we performed the linking of peaks to genes individually in nuclei from CocUD and Ctrl individuals. We then performed an overlap analysis of the identified links, filtered for absolute correlation coefficients > 0.1 , and grouped the remaining links into three categories: CocUD-specific: peak-gene associations only identified in CoCUD, Ctrl-specific: peak-gene associations only identified in Ctrl and common: peak-gene associations identified in both, CocUD and Ctrl.

Comment 3.5: It is not clear the relationship between Figure 3A and Figure 3E?

Response 3.5: We thank the reviewer for this comment. Figure 3E depicts the results of the differential motif activity using chromVAR. In the figure legend we did not label this sufficiently. We thus corrected the figure legend which now states:

Figure 3. A) Heatmap depicting shared (brown), CocUD-specific (red), and control-specific (blue) links between chromatin accessibility and RNA expression by cell type. B) emapplot showing overrepresented gene ontology biological processes for genes characterized by a CocUD-specific linked peak in A), color represents statistical significance and circle size the number of genes in the GO term. C) heatmap depicting cell type-specific motif enrichment, yellow motif names indicate increased activity while motif names in blue represent reduced activity of the motif in cell types based on chromVAR activity scoring. D) Feature plots confirming motif expression for MA1114.1/PBX3 in D1- and D2-MSNs, MA1577.1/TLX2 in astrocytes and MA0080.5/SPI1 in Microglia. Enrichment scores are color-coded from blue (no enrichment) to yellow (strong enrichment). E) Heatmap depicting differential transcription factor motif activity between CocUD and controls, red = increased activity in CocUD, blue = decreased activity in CocUD. F) Heatmap depicting corresponding transcription factor differential transcript expression in CocUD, red = upregulated in CocUD, blue = downregulated in CocUD. G) transcription factor motifs for ZEB1 and FOXP2, size of sequence represents probability. H) feature plots depicting ZEB1 and FOXP2 expression in cell types of the caudate nucleus (log-normalized counts).

Comment 3.6: In Fig4D, why did the author not use the double delta CT method?

Response 3.6: We thank the reviewer for this comment that was also raised by reviewer 1. We now use the $2^{-\Delta\Delta C}$ method (see response 1.4).

Comment 3.7: the description of MOFA is not clear. How do the factors identified in this analysis connect to CoCUD besides some similar enriched GO terms? How do we explain the negative enrichment ZEB1 in factor 8 gene set?

Response 3.7: We thank the reviewer for this comment and have added a more detailed description to the MOFA section. We used MOFA as a complementary and confirmatory approach to the individual DE analyses in RNA and ATAC datasets to better cover RNA-ATAC interactions in our multi-omics dataset. The MOFA factors are correlated with CocUD status, as shown in the Heatmap in Figure S4A. For example, Factor8 is positively associated with CocUD status, it has a negative motif enrichment for ZEB1, which is in line with the negative motif enrichment in the ChromVAR analyses and the negative log2FC in the differential expression analysis. While GO enrichment analyses on DE genes are useful to further investigate the role of CocUD-associated genes, a GO-enrichment analysis on MOFA factor loadings allows us to investigate CocUD-associated genes that show the multi-modal relationship with ATAC signals. Thus, as the approaches are intrinsically different, we used them to identify convergent evidence at the pathway level, which we found, for instance, for neuronal and metabolic pathways.

Line 263-282:

Multi-Omics Factor Analysis reveals a sex-specific factor characterized by metabolic pathway enrichment

In addition to the data-informed approaches, we performed a data-driven multi-omics factor analysis to identify a dimensionality-reduced set of factors associated with CocUD. MOFA factors capture both RNA and ATAC information and are therefore particularly informative about multi-modal (i.e. multi-omics) interactions between RNA and ATAC signals. Of the 15 resulting factors, four were associated with CocUD (Supplementary Fig. 4A and 4B). Factor5 is

characterized by high factor loadings for *GABAergic CALB2* and *PTHLH PVALB* cells, with positive enrichment for *synaptic signaling and neurodevelopment* and negative enrichment for pathways related to *synapse maturation and gliogenesis* (Supplementary Fig. 4B & 4C).

Factor7 showed a *CocUD-specific association with D2-MSNs* and was enriched for GO terms related to *the ribosome and translation pathways* (Supplementary Fig. 4D). Overall, the results for the *CocUD-associated factors confirmed the findings of the DE analysis*.

Of particular interest is the *CocUD-associated Factor8* that indicated sex-specific subclusters in the majority of cell types, marked by high factor-loadings in subclusters of *OPCs, oligodendrocytes, and particularly D1- and D2-MSNs*. Characterized by an enrichment for metabolic GO terms, this factor suggests a male-specific effect. Interestingly, in the promoter-associated motifs of Factor8, we observe a negative enrichment of major regulators from the GRN, such as *ZEB1 and TCF4* (Supplementary Fig. 4E), also in line with the *DE and chromVAR analyses*.

Comment 3.8: The descriptions of sex-DE analysis and GWAS enrichment are too simple to understand. Especially for negative results of GWAS enrichment at the cell type level.

Response 3.8: We thank the reviewer for this comment and now provide more details about sex-specific and GWAS enrichment analysis:

In the RRHO figure (Fig. 4E), we have added arrows to indicate which panels indicate concordance/divergence of expression patterns across sexes.

Further, we have modified the results section:

Sex-stratified DE analysis shows concordant effects in astrocytes, microglia, and D1-/D2-MSNs (line 284-294)

Because we observed high Factor8 loadings in the majority of cell types, we performed exploratory analyses on potential sex-specific transcriptional signatures of CocUD. Using rank-rank hypergeometric overlap (RRHO) analysis, we aimed to identify concordant and discordant CocUD-associated transcriptional changes in individual cell types. Interestingly, we observed high concordance of CocUD-associated DE patterns between sexes in astrocytes, GABAergic D1MSN, and microglia, and moderate concordance between sexes in GABAergic D2 MSNs, and OPCs (Fig. 4E). Strong divergent patterns were observed in GABAergic CALB2 and

oligodendrocyte clusters suggesting sex-specific transcriptional changes in these clusters which further underscores the results from MOFA.

No cell type-specific enrichment of GWAS signals of CocUD (line 296-306)

To test whether GWAS signals of substance use disorders and CocUD in particular, were overrepresented in cell type gene expression patterns, we used single-cell disease-relevance scoring (scDRS). In scDRS, a polygenic enrichment analysis is performed for each cell to evaluate whether its expression profile is significantly enriched for phenotype-associated risk genes. In microglia, oligodendrocytes, and cholinergic neuron clusters, we found the strongest CocUD-associated enrichment patterns in individual cells similar to the results observed for alcohol dependence (Supplementary Fig. 4F). However, we did not observe statistically significant enrichment for any of the investigated SUDs, either at the single cell or the cluster level, most likely due to the small GWAS sample sizes in AA populations (Supplementary Table 9, Methods section).

Response Letter - Cell type-specific Multi-Omics Analysis of Cocaine Use Disorder in the Human Caudate Nucleus

Original reviewer comments are in **bold**.

Our responses are in *italics*.

REVIEWERS' COMMENTS

Reviewer #2 (Remarks to the Author):

Previous concerns were adequately addressed. I have no further comments at this time.

Reviewer #2 (Remarks on code availability):

N/A

Reviewer #3 (Remarks to the Author):

I appreciate authors addressed my comments, I don't have more concerns.

We thank the reviewers for the constructive comments in the revision and are happy that all comments and concerns have been addressed.

Reviewer #3 (Remarks on code availability):

A detailed readme on GitHub should be available for the deposited files.

We included a detailed readme on the GitHub page.

Cell type-specific Multi-Omics Analysis of Cocaine Use Disorder in the Human Caudate Nucleus (Zillich et al., 2025)

- 1_initial_QC - QC scripts per sample
- 2_merge_call_peaks - seurat files are merged and a joint peak calling is performed (MACS2)
- 3a_RNA - Preprocessing of RNA slot
- 3b_ATAC_RNA_integration - Preprocessing of ATAC slot, integration with RNA slot using harmony
- 4_Markers_Integrated_Figure1_S5 - Cluster annotation, code to produce Figure 1 and Supplementary Figure 5
- 5a_MOFA_prep - create MOFA object, run MOFA
- 5b_MOFA_downstream_FigureS4 - downstream analysis of MOFA, code to produce Supplementary Figure 4A-4E
- 6_DE_DA_analysis_Figure2_FigureS2 - differential expression and accessibility analysis, code to produce Figures 2 and Supplementary Figure 2
- 7_motifs_TF_Figure3 - motif analyses, code to produce Figure 3
- 8_Pando_MSN_Figure4_S3 - pando in D1/D2 medium spiny neurons, code to produce Figure 4A-4C and Supplementary Figure 3
- 9a_scDRS_FigureS4 - scDRS analysis
- 9b_visualize_scDRS_FigureS4 - visualisation of scDRS results, code to produce Supplementary Figure 4F
- 10_RRHO_sex_specific.Rmd - sex-specific DE and RRHO analysis, code to produce Figure 4E
- 11_qPCR.R - analysis of qPCR results, code to produce Figure 4D
- 12_tables_manuscript.R - code to produce Supplementary Tables